# To Steer or Not to Steer?
# Mechanistic Error Reduction with Abstention for Language Models

**Anna Hedström** [1]  **Salim I. Amoukou** [2]  **Tom Bewley** [2]  **Saumitra Mishra** [2]  **Manuela Veloso** [2]

## Abstract

We introduce **Mechanistic Error Reduction with Abstention** (**MERA**), a principled framework for steering language models (LMs) to mitigate errors through selective, adaptive interventions. Unlike existing methods that rely on fixed, manually tuned steering strengths, often resulting in under or oversteering, **MERA** addresses these limitations by (i) optimising the intervention direction, and (ii) calibrating *when*, and *how much* to steer, thereby provably improving performance or abstaining when no confident correction is possible. Experiments across diverse datasets, and LM families demonstrate safe, effective, non-degrading error correction, and that **MERA** outperforms existing baselines. Moreover, **MERA** can be applied on top of existing steering techniques to further enhance their performance, establishing it as a general-purpose, and efficient approach to mechanistic activation steering.

## 1. Introduction

Despite the impressive capabilities of current language models (LMs), they can be frustratingly error-prone. Failures arise not only in open-ended tasks like reasoning, factual consistency or planning (Kambhampati et al., 2024), but also in simple prediction settings (Hendrycks et al., 2021b; Webson & Pavlick, 2022; Zhou et al., 2024). Mitigating such errors is an open research question.

Error reduction methods for LMs have evolved in several directions, including fine-tuning (Hu et al., 2022; Wu et al., 2024; Yin et al., 2024), and inference-time approaches like prompt engineering (Kojima et al., 2022), and guided decoding (Yang & Klein, 2021; Liu et al., 2021). While effective

[1]Work done during an internship at J.P. Morgan AI Research. [2]J.P. Morgan AI Research. Correspondence to: Anna Hedström <hedstroem.anna@gmail.com>, Salim I. Amoukou <salim.ibrahimamoukou@jpmorgan.com>.

*Proceedings of the $42^{nd}$ International Conference on Machine Learning*, Vancouver, Canada. PMLR 267, 2025. Copyright 2025 by the author(s).

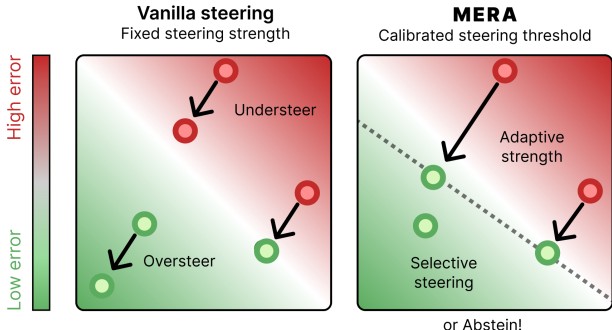

*Figure 1.* An illustration of traditional *vanilla steering* with fixed steering strengths (left), leading to under-, and oversteering, and **MERA** (right), providing calibrated steering thresholds.

for specific goals, these approaches are typically computationally intensive or context-sensitive.

An alternative approach is to intervene in the internal computations of the model itself. Mechanistic steering (or "representation engineering") has emerged as a promising line of research to influence model behaviour by intervening on activations in inference-time, without permanent weight updates (Li et al., 2023; Belrose et al., 2023; Todd et al., 2024; Qiu et al., 2024; Turner et al., 2024).

One widely used instantiation of this approach is *contrastive steering* (Arditi et al., 2024; Rimsky et al., 2024), where a steering vector is constructed by taking the mean activation difference between paired examples with, and without a targeted concept (*e.g.* toxic vs non-toxic outputs). This vector can then be added to the model's activations to amplify or suppress such concept in the model (Ball et al., 2024), under the assumption that the concept is represented along a linear direction in the model's representation space (Elhage et al., 2022; Marks & Tegmark, 2024; Park et al., 2024).

A natural extension is to explore whether such *additive linear* steering can be applied for the specific goal of *error mitigation*: directly reducing model errors on prediction tasks. That is, can we find a linear direction in LM's activation space that encodes the concept of "being wrong", and then steer away from it? Unlike traditional alignment tasks such as reducing toxicity or harmful responses, steer-

ing for error mitigation may be more challenging (Engels et al., 2024). Errors are not necessarily tied to a single identifiable concept, but can manifest in various forms (Wang et al., 2025; Orgad et al., 2024), rendering them resistant to additive steering (Belrose et al., 2023).

An emerging line of research explores alternative steering techniques for error-related properties like truthfulness, and hallucination (Li et al., 2023; Qiu et al., 2024; Wang et al., 2025; Bhattacharjee et al., 2024). They involve complex departures from the elegant simplicity of additive steering through the addition of clustering, architecture-specific interventions, or label-based contrastive vectors. Rather than advancing into such techniques, we address a simpler, and more fundamental question: *when, and how much should we steer to mitigate errors effectively?*

As Figure 1 (left) illustrates, traditional steering methods rely on fixed steering strengths. This can lead to issues such as understeering, where an insufficient change is made to mitigate errors, and oversteering, where unnecessary (and potentially detrimental) steering is applied (Tan et al., 2024; Scalena et al., 2024). The dominant practice is to empirically find a fixed steering strength based on ad hoc hyperparameter sweeps (Liu et al., 2024; Turner et al., 2024; Postmus & Abreu, 2024; Lee et al., 2025), which are costly, model-specific, and lack generalisation. Others apply fixed strengths or none at all (Arditi et al., 2024; Durmus et al., 2024; Ball et al., 2024). There are also conditional approaches (Wang et al., 2025; Scalena et al., 2024; Cheng et al., 2024) that adjust the steering strength dynamically, but these are not adapted for the specific objective of error mitigation (see further discussion in §A.3).

In this work, we address the challenge of steering strength calibration for error mitigation. After identifying steering directions using linear error estimation probes instead of contrastive pairs (see §4), we show how to derive an optimal steering strength to reduce empirical error. This results in a steering method that does not use a fixed strength, but rather a fixed *threshold* in the activation space to which steering is applied; see Figure 1 (right). A by-product of our approach is that we *abstain* from steering activations that are beyond the threshold, for which error is already predicted to be low. By formulating steering as a constrained optimisation problem, we derive a closed-form solution (see §3.1) which ensures that interventions grow proportionally with the predicted error. The following calibration step (see §3.2), checks per task whether a threshold exists that *confidently* improves performance; if none can be found, we abstain entirely. In this way, we steer only when it is provably beneficial, effectively addressing both under-, and oversteering.

We make the following contributions:

**C1** We propose a principled steering framework to mitigate errors, which calibrates intervention intensity to prevent both under-, and oversteeering, guaranteeing improved or non-degrading performance (see §3).

**C2** We investigate which representation spaces, sparse autoencoder (SAE) features or original activations, are most effective for identifying error-mitigation directions in LMs using linear probes (see §4).

**C3** We introduce **mechanistic error reduction with abstention** (**MERA**, see §5), a practical method that empirically achieves safe, effective, non-degrading error correction, and improve upon existing baselines across a range of models, and tasks (see §6).

This work highlights a broader vision for post-training alignment: lightweight interventions that are both effective, and probabilistically safe by design. While our current focus is error mitigation, by adapting the target signal with labeled inputs, **MERA** could steer LMs toward diverse *specialised* objectives (e.g., harmlessness, honesty, and fairness etc). This positions **MERA** as a general-purpose framework for principled, and minimal post-hoc model control.

## 2. Preliminaries

Consider an autoregressive, decoder-only transformer language model $f$ that maps $n$ input tokens, $\mathbf{x} = (x_1, \ldots, x_n) \in \mathbb{R}^n$, to an output probability distribution over all input, and generated tokens, $\mathbf{a} = (a_1, \ldots, a_{m+n}) \in \mathbb{R}^{(m+n) \times |\mathcal{V}|}$, where $\mathcal{V}$ is the vocabulary set. The model consists of $L$ sequential blocks, each comprising attention, feed-forward, and residual stream layers. At layer $\ell \in [L]$, the activation for the token at position $i$ is denoted $h_i^{(\ell)}(\mathbf{x})$. For each input $\mathbf{x}$, and a given task, we assume an error function $E(\mathbf{a}) \in [0, 1]$ that measures the quality of the model's generated output, where lower is better.

This setup is general, and applies to various LM tasks. For instance, in multiple-choice question answering (MCQA), $E(\mathbf{a})$ can be 0 if the correct answer is within the generated tokens, and 1 otherwise. In summarisation, $E(\mathbf{a})$ can quantify the distance between the generated summary, and a reference.

### 2.1. LMs for Supervised Tasks

In this work, we focus on supervised problems, where each prompt $\mathbf{x}$ is associated with a ground truth label $y \in \mathcal{V}$. Let $\mathcal{Y} \subset \mathcal{V}$ be the set of valid labels (see §A.5 for details), and $\text{idx}(y) \in \{1, \ldots, |\mathcal{V}|\}$ be the index of label $y \in \mathcal{V}$ in the vocabulary. Given the model's output distribution $\mathbf{a}$, we select a token position $k$, and parse a predicted label $\hat{y}$, and

its probability $\text{prob}_{\hat{y}}$ as follows

$$\hat{y} = \arg\max_{y \in \mathcal{Y}} \mathbf{a}_{k,\text{idx}(y)} \quad \text{and} \quad \text{prob}_{\hat{y}} = \frac{\mathbf{a}_{k,\text{idx}(\hat{y})}}{\sum_{y \in \mathcal{Y}} \mathbf{a}_{k,\text{idx}(y)}}.$$

We choose a token position $k$ using one of two strategies: (i) *last*, which uses the final token $\mathbf{a}_n$ in the input sequence; or (ii) *exact*, a recently proposed method (Orgad et al., 2024) that uses the first token that matches any valid ground truth label. In supervised LM tasks, it is common to interpret the logits at the final token position as the model's prediction, as this captures the model's decision after processing the full prompt. However, the last position does not always correlate with how the model actually generates answers (Pres et al., 2024). To address this, the *exact* position complements the *last* prediction mode, selecting the first token that matches any valid ground truth label, measuring how an LM model behaves in its free-form, open-ended generated response. Further experimental details are provided in §A.5.

For a given prompt-label pair $(\mathbf{x}, y)$, the error function is

$$E(\mathbf{a}) = 1 - \text{prob}_y, \tag{1}$$

where $\text{prob}_y$ is evaluated at the chosen token position $k$. Additionally, we assess the model's end-task performance, which, in our supervised setting, corresponds to accuracy, which is defined as

$$A(\mathbf{a}) = \mathbb{1}[\hat{y} = y]. \tag{2}$$

## 2.2. Steering

In this paper, we focus on the concept of *additive steering* (or "activation addition"). Suppose $E(\mathbf{a}) \in \{0, 1\}$, where $E(\mathbf{a}) = 1$ indicates that the generated output exhibits a certain targeted concept (*e.g.* refusing harmful instructions). Steering aims to add a vector $v_i^{(\ell)}(\mathbf{x})$ to the activations $h$ at one or more token positions, and layers as follows

$$\tilde{h}_i^{(\ell)}(\mathbf{x}) = h_i^{(\ell)}(\mathbf{x}) + \lambda \, v_i^{(\ell)}(\mathbf{x}), \tag{3}$$

such that after this intervention, the new output $\tilde{\mathbf{a}}$ satisfies $E(\tilde{\mathbf{a}}) = 0$ (*e.g.* refusing harmful content). A scalar parameter $\lambda \in \mathbb{R}^+$ is introduced to control the strength of the steering. Throughout the paper, we use the tilde notation $\tilde{\cdot}$ to represent the steered version of any quantity, and let $h := h_i^{(\ell)}(\mathbf{x})$, and $v := v_i^{(\ell)}(\mathbf{x})$ for brevity. We denote $\tilde{f}$ the steered model with output $\tilde{\mathbf{a}} := \tilde{f}(\mathbf{x})$. Several methods have been proposed to extract steering directions $v$, including contrastive pairs, linear probes, and principal components (Wu et al., 2025). In this work, we focus on the first two, which can be interpreted as deriving $v$ from a classifier of the targeted concept (Mallen et al., 2023).

**Contrastive Steering** A widely used technique (Zou et al., 2023; Arditi et al., 2024; Ball et al., 2024; Rimsky et al., 2024; Farquhar et al., 2023; Marks & Tegmark, 2024), involves intervening across a token position $i$, and layer $\ell$ by defining $v_i^{(\ell)}(\mathbf{x})$ as the *difference-in-means* between activation values of positive examples ($E(\mathbf{a}) = 1$), and negative examples ($E(\mathbf{a}) = 0$). Given a dataset $\mathcal{D} = \{(\mathbf{x}_j, y_j)\}_{j=1}^D$, we define the set of positive examples as $\mathcal{D}_+ = \{j \in [D] : E(\mathbf{a}_j) = 1\}$, and the set of negative examples as $\mathcal{D}_- = \{j \in [D] : E(\mathbf{a}_j) = 0\}$, where $\mathbf{a}_j = f(\mathbf{x}_j)$. The steering vector $v(\mathbf{x})$ is then defined as

$$v(\mathbf{x}) = \mu_+ - \mu_-, \tag{4}$$

where the mean activations $\mu_+$, and $\mu_-$ are given by

$$\mu_+ = \frac{1}{|\mathcal{D}_+|} \sum_{j \in \mathcal{D}_+} h(\mathbf{x}_j), \; \mu_- = \frac{1}{|\mathcal{D}_-|} \sum_{j \in \mathcal{D}_-} h(\mathbf{x}_j),$$

which is computed for each layer $\ell$, and token position $i$ as $h := h_i^{(\ell)}(\mathbf{x})$, and $v := v_i^{(\ell)}(\mathbf{x})$.

**Probe-based Steering** An emerging alternative technique is steering guided by probes (von Rütte et al., 2024; Cheng et al., 2024), which leverages learned linear classifiers (or "probes") to guide intervention. A probe, $\hat{p}(h) = w^\top h$, is trained to classify targeted properties, and the classifier weights $w$ are directly used as the steering vector $v$. Contrastive steering can be viewed as a special case of this framework, where the steering vector is derived from the Linear Discriminant Analysis classifier, assuming isotropic covariance between $\mu_+$, and $\mu_-$ (Mallen et al., 2023).

**Goal** This paper focuses on error mitigation through steering by leveraging the interplay between probes, and activation adjustments. In what follows, we propose a principled framework for utilising probe models in steering, addressing key challenges such as determining optimal steering strength $\lambda$, and adaptively intervening across layers, and token positions with the ability to abstain.

## 3. Conditional Steering for Error Mitigation

We extend probe steering to minimise continuous errors, such as the model's error $E(\mathbf{a})$. Instead of training a classifier, we train an error estimator $\hat{p}(h)$ to predict $E(\mathbf{a})$ from activations $h$.

In prior work (Mallen et al., 2023; Li et al., 2023; von Rütte et al., 2024), the probe's weights were used directly as the steering vector. In contrast, we propose to use both the probe's *predictions*, and its weights. Since the probe's predictions estimate the LM's error, we construct the steering vector to explicitly reduce the predicted error. This approach provides a principled mechanism for determining the steering strength, as demonstrated in the next section.

## 3.1. Optimising Steering with Abstention

We define steering as moving the activation $h$ in a direction that reduces the predicted error while staying close to the original $h$. Formally, we solve the following constained optimisation problem

$$\min_{v} \ \|v\|_2^2 \quad \text{subject to} \quad \hat{p}(h + v) \leq \alpha, \qquad (5)$$

where $\alpha$ represents the target threshold of error reduction.

**Linear Case** If $\hat{p}(h)$ is linear, *i.e.* $\hat{p}(h) = w^\top h$, this optimisation problem admits a closed-form solution as

$$v^\star = \begin{cases} 0, & \text{if } w^\top h \leq \alpha, \\ \left( \dfrac{\alpha - w^\top h}{\|w\|_2^2} \right) w, & \text{if } w^\top h > \alpha. \end{cases} \qquad (6)$$

Intuitively, if $w^\top h$ is already at or below $\alpha$, no intervention is required. Otherwise, we adjust $h$ to ensure $w^\top(h + v^\star)$ satisfies the threshold $\alpha$.

Equivalently, we can rewrite the resulting steering vector as $v^\star = \lambda^\star w$, where

$$\lambda^\star = \max \left( 0, \ \frac{\alpha - w^\top h}{\|w\|_2^2} \right). \qquad (7)$$

This formulation offers intuitive interpretations:

- **Selective Steering:** Intervention are applied selectively, only when $\hat{p}(h) = w^\top h > \alpha$ at a given token, and layer level.

- **Adaptive Strength:** The steering strength scales with the residual $\alpha - w^\top h = \alpha - \hat{p}(h)$, leading to stronger adjustments for larger estimated errors.

The normalisation by $\|w\|_2^2$ aligns with prior work (von Rütte et al., 2024), where such scaling factors have been shown to improve steering efficacy.

**Extension to Non-linear Models** A similar closed-form solution arises when $\hat{p}(h)$ is an affine function composed of an invertible transformation. For instance, if $\hat{p}(h) = \text{sigmoid}(w^\top h)$, the constraint $\hat{p}(h + v) \leq \alpha$ translates to $w^\top(h + v) \leq \text{logit}(\alpha)$. The steering strength will be

$$\lambda^\star = \max \left( 0, \ \frac{\text{logit}(\alpha) - \hat{p}(h)}{\|w\|_2^2} \right) \qquad (8)$$

Although we focus on linear probes in this work, §A.1 discusses how to extend these ideas to non-linear probe functions (*e.g.* a one-layer MLP).

## 3.2. Calibrating for Safety

In this framework, the steering threshold $\alpha$ is the key parameter that determines both *when* an intervention is triggered, and *how strongly* the activations are shifted. We select $\alpha$ using a calibration set $\mathcal{D}_{\text{cal}} = \{(\mathbf{x}_j, y_j)\}_{j=1}^N$, which is distinct from the dataset used to train the error regressor. For a given $\alpha$, let $\Delta_{\text{cal}}(\alpha)$ denote the change in some measure of LM's performance on $\mathcal{D}_{\text{cal}}$, such as the accuracy $A$ (Equation 2) or (negative) error $-E$ (Equation 1), after applying steering with the corresponding steering strength $\lambda^\star$ (Equation 7). Specifically, we define the optimal threshold $\alpha^*$ as

$$\alpha^* = \arg \sup_{\alpha \in \alpha_{\text{valid}}} \Delta_{\text{cal}}(\alpha), \ \text{where}$$

$\alpha_{\text{valid}} = \{\alpha \in \{\alpha_1, \dots, \alpha_K\} : \Delta_{\text{cal}}(\alpha) > \epsilon + b(\delta, K, N)\}$, and $\alpha_1, \dots, \alpha_K \in (0, 1)$ are candidates values for $\alpha$. $b(\delta, K, N) = \sqrt{\log(2K/\delta)/(2N)}$ is a confidence bound derived by applying a union bound (Bonferroni correction, using a Hoeffding inequality) over the $K$ candidate hypotheses $\alpha_i$ (see §A.2 for derivation). This ensures that, with probability at least $1 - \delta$, the selected $\alpha^*$ yields a genuine performance improvement that exceeds $\epsilon$.

**Theoretical Guarantees** This procedure guarantees either a provable improvement in performance or abstains from intervention if $\alpha_{\text{valid}}$ is empty. Assuming i.i.d. samples, it satisfies

$$\mathbb{P}\left( \Delta_{\text{cal}}(\alpha^*) > \epsilon \right) \geq 1 - \delta. \qquad (9)$$

See proof in §A.2. More generally, this approach is flexible, and can incorporate any valid bound $b$ satisfying the desired confidence guarantee in Equation 9.

Instead of relying on a Bonferroni-style correction, which may be overly conservative, an alternative approach is to split the calibration data: use one part to select the optimal $\alpha^\star$, and the other to verify that its confidence interval lower bound exceeds $\epsilon$.

By adjusting $\delta$, practitioners can control the trade-off between conservativeness, and aggressiveness of the steering policy. In our experiments, we set $\delta = 0.01, \epsilon = 0$, and use difference in accuracy as $\Delta_{\text{cal}}(\alpha)$, though the method is agnostic to the metric used, and can accommodate relevant alternatives such as F1 score in classification tasks.

## 3.3. A Principled Framework

Our framework establishes a principled foundation for *conditional* activation steering, grounded in three key ingredients. First, we use linear probes to obtain effective directions for minimising the predicted error. Second, this direction is then scaled using the closed-form solution at both the token, and layer levels. Third, we calibrate the steering threshold $\alpha$ against the true error on the calibration dataset, informed by the user's tolerance for uncertainty (*i.e.* set by $\delta$).

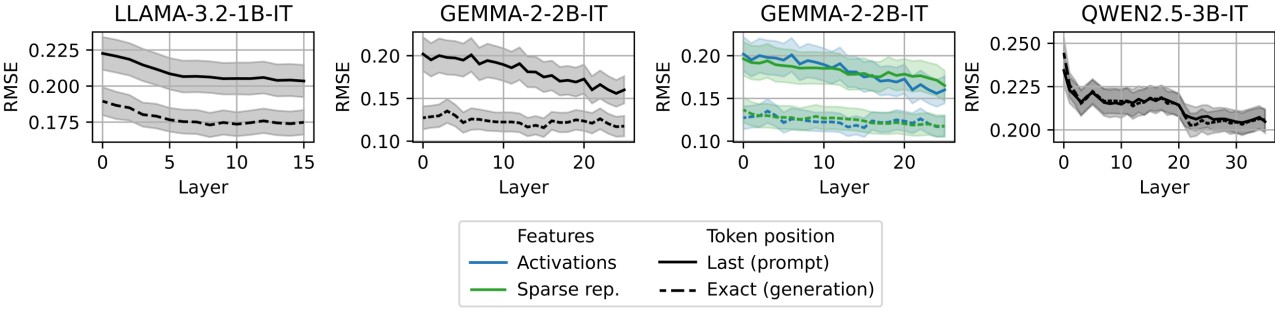

*Figure 2.* Layer-wise performance of linear error estimators for different LM families, separated by distinct token position, and input feature strategies. RMSE, and standard error is reported. Extended results are provided in A.6.1.

A key advantage of this formulation lies in how it handles the steering strength $\lambda$: defining the appropriate range of values for this factor is nontrivial, and often requires careful weight normalisation or manual tuning. In our approach, this challenge is eliminated. The $\alpha$ value, and the estimated error inherently determine the steering strength, providing a natural, and intuitive interpretation of its role:

- **Global Abstention.** If calibration indicates that no $\alpha$ offers improvement beyond a baseline $\epsilon$, we refrain from intervening altogether.

In this way, steering interventions remain both effective (reducing error where needed), and safe (avoiding unnecessary or detrimental modifications). In a similar spirit to Cheng et al. (2024); Liu et al. (2024), which underscores the value of theoretical guarantees for safe steering towards semantic, binary properties, our method, to the best of our knowledge, is the first error-mitigation approach whose objective is to *calibrate* steering thresholds to lower true empirical error with provable guarantees, thereby advancing this emerging line of research.

## 4. Choosing Representations for Steering

In the previous section, we showed how to derive a conditional steering direction for error mitigation from an arbitrary model activation $h$. A remaining question is: *which representations should be used to find steering directions?*

The candidate set of activations $h$ is large, spanning $(n + m) \times L$, where $n + m$ denotes the total number of tokens from both the prompt, and the generated answer, and $L$ is the number of layers. Existing studies on contrastive steering navigate this extensive space by conducting computationally expensive, and model-specific hyperparameter sweeps (Li et al., 2023; Liu et al., 2024; Turner et al., 2024). In addition, the strategies used for selecting layers, and token positions for constructing the steering vector versus applying the vec-

tor in inference-time, often differ (see §A.4 for an extended discussion). Lastly, it remains an open research question whether such strategies extend to mitigating *continuous* errors (as opposed to strictly binary alignment targets, *e.g.* reducing toxicity).

To address these gaps, we investigate two central questions:

- **Q1 Token position.** Does extracting activations from the *exact* token position in the generated answer, as recently recommended by Orgad et al. (2024), improve probe performance?

- **Q2 Representation.** Can *sparse* representations, such as those obtained through SAEs (Lieberum et al., 2024; Joseph Bloom & Chanin, 2024), enhance the learned error direction, and thus improve steering performance?

In line with prior work (Rimsky et al., 2024; Ball et al., 2024; Tan et al., 2024; Arditi et al., 2024; Jorgensen et al., 2023; Krasheninnikov & Krueger, 2024; Postmus & Abreu, 2024), we focus on steering the *residual stream* (Elhage et al., 2021), *i.e.* the aggregated "running sum" representation across MLP, and attention layers (Zhao et al., 2021), rather than selecting specific attention heads (Li et al., 2023; Wang et al., 2025).

### 4.1. Experimental Setup

We systematically analyse how choices of token position, and representation type affect probe quality across multiple models, and tasks.

**Models**  We evaluate a diverse set of decoder-only models, including LLaMA (Team, 2024b), GEMMA (Team, 2024a), and QWEN (Team, 2024c), spanning distinct sizes. Specifically, we examine both the base-, and instruction-tuned variants of LLaMA-3.2-1B, GEMMA-2-2B, and QWEN-2.5-3B models. Details on these LMs, and their task performance are provided in §A.5. For the Gemma models, we

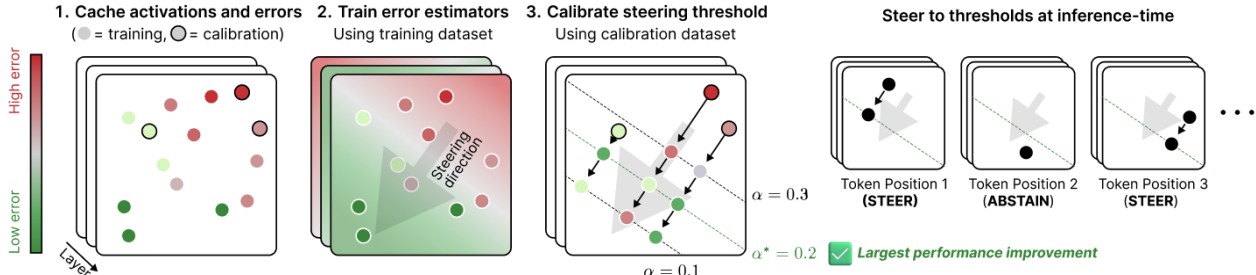

*Figure 3.* Visualisation of **MERA** methodology to mechanistically steer LMs: **1.** cache activations, and errors, **2.** train error estimators, **3.** calibrate steering thresholds.

retrieve pre-trained SAEs from GEMMA-SCOPE (Lieberum et al., 2024).

**Datasets** We evaluate our methods on supervised tasks with varying label cardinalities (binary, ternary, and quaternary), and class distributions, ranging from balanced to highly imbalanced datasets. The datasets include SMS SPAM (Almeida et al., 2011), YES/NO (AI, 2023), SENTIMENT (AI, 2023), and MMLU (Hendrycks et al., 2021a), with subset of high-school, and professional level questions, *i.e.* MMLU-HS, and MMLU-PROF. The training datasets contain between 2600-300 training samples, with 30% used for validation. Additional dataset details are provided in Table 2 in §A.5.

**Probes** For each layer, we train a linear probe $\hat{p}$, and measure how well it predicts the model's true error $E(\mathbf{a})$ across distinct token positions. Regularisation ensures that the learned error estimators are sparse (see §A.6.3 for further details). We use root mean squared error (RMSE) to measure performance, where lower values indicate better performance (see §A.6.4 for details).

**Sparse Representations** For sparse representations, we specifically explore the encoding of a SAE, which maps the original activations $h \in \mathbb{R}^d$ to a high-dimensional activation space $z(h) \in \mathbb{R}^w$ as follows

$$z(h) = \sigma(W_{\text{enc}}(h - b_{\text{enc}}) + b_{\text{enc}}),$$

where $\sigma$ is an activation function (*e.g.* JumpReLU (Lieberum et al., 2024)), $b_{\text{enc}}$ is a bias vector, and $W_{\text{enc}}$ is a learned weight matrix. Only a subset of activations in $z$ will be non-zero, and $w \gg d$. Empirically, we explore whether sparse activations enhance linear probe performance (Templeton et al., 2024) to potentially be used for steering (Zhao et al., 2024; Durmus et al., 2024; Mayne et al., 2024; Bricken et al., 2024; Kantamneni et al., 2025).

**Results** Figure 2 illustrates the RMSE, and standard error across the top 5 performing regression probes (see §A.5 provides additional details). Generally, we find that the exact position strategy performs at least as well as, if not better than, the last position across most model families (*i.e.* LLAMA, and GEMMA). Thus, we will use the exact position to extract $h$ to train $\hat{p}$. For sparse representations, we find no signal of improved probe performance. In §A.6.1, we break down the results per model, and observe a high variability across models. Since SAEs also impose high compute costs, we find no compelling reason to use sparse activations instead of original activations for the purpose of probe-based steering.

## 5. Introducing MERA

In the following, we introduce **MERA**—a general mechanistic steering methodology that practically operates in three main steps. We refer to Figure 3 for an overview of these steps.

1. **Cache activations, and errors.** Construct a training $\mathcal{D}_{\text{train}}$ by pairing LM errors $E(\mathbf{a})$ with activations $h_k^{(\ell)}$ at exact token position $k$ for each layer $\ell \in \{1, \ldots, L\}$. Prepare a calibration dataset $\mathcal{D}_{\text{cal}}$ with input prompts $\mathbf{x}$, and their corresponding true labels $y$.

2. **Train error estimators.** Train $L$ linear probes $\hat{p}(h) = w^\top h$ on $\mathcal{D}_{\text{train}}$ activations $h$ to estimate the LM error $E(\mathbf{a})$, applying distinct sparsity constraints (see full training methodology in §A.5).

3. **Calibrate steering threshold.** Calibrate the optimal steering threshold $\alpha^*$ on inputs $\mathbf{x}$, and their corresponding true labels $y$ in $\mathcal{D}_{\text{cal}}$ to maximise $\Delta_{\text{cal}}(\alpha)$ (see §3.2), given a user-specified confidence $1 - \delta$.

A key advantage of **MERA** is its flexibility: while we train linear error estimators to find a steering direction $v$ for each layer $\ell$, both the optimisation of $\lambda$, and calibration of $\alpha$

remains generalisable to other definitions of $v$, such as contrastive steering (Farquhar et al., 2023) or weights from a logistic regression probes (Cheng et al., 2024; von Rütte et al., 2024). This means we can use **MERA** as a *methodology* to improve existing steering methods. In §6, we empirically quantify this improvement.

**How we Calibrate** We determine the optimal steering threshold $\alpha^\star$ via a gradient-free grid search over $\alpha \in [0, 1]$ discretised into 10 equal intervals, selecting the value that maximises $\Delta_{\text{cal}}(\alpha)$ while satisfying a predefined safety constraint as described in §3.2. The unsteered reference accuracy is first established on $\mathcal{D}_{\text{cal}}$, then accuracy is reported when steering at each candidate $\alpha$. If no $\alpha$ yields a statistically significant improvement, we abstain from intervention, ensuring $\lambda^\star = 0$. In §A.7, we show that accuracy, and error are closely related in our tasks. As discussed in §A.5, **MERA** offers efficient inference, with the main computational burden arising from the one-time, offline calibration.

# 6. Benchmarking

Our experiments aim to answer the following question:

**Q1** Does **MERA** make steering more effective, and safe?

## 6.1. Evaluating Steering

In alignment tasks, steering is typically evaluated along two complementary dimensions, *i.e.* its efficiency in producing responses aligned with the high-level concept, and its ability to preserve fluency, and naturalness of text (Pres et al., 2024). In the context of error mitigation, however, the goal is to steer LMs toward correct labels. Accordingly, we shift the evaluation target to improving *task accuracy*.

Our main question is: how much does a steering method actually help (or hurt) model performance? Comparing raw performance deltas alone can be misleading, especially when baseline accuracy is already high, making small degradations disproportionately large. To address this, in the spirit of Burns et al. (2023), we define a *Steering Performance Impact (SPI)* score

$$\mathbf{SPI} = \begin{cases} \frac{\tilde{A}_{\mathcal{D}_{\text{test}}} - A_{\mathcal{D}_{\text{test}}}}{1 - A_{\mathcal{D}_{\text{test}}}}, & \text{if } \tilde{A}_{\mathcal{D}_{\text{test}}} > A_{\mathcal{D}_{\text{test}}} \\ \frac{\tilde{A}_{\mathcal{D}_{\text{test}}} - A_{\mathcal{D}_{\text{test}}}}{A_{\mathcal{D}_{\text{test}}}}, & \text{otherwise} \end{cases} \quad (10)$$

where $A_{\mathcal{D}_{\text{test}}}$ is the test set accuracy $\mathcal{D}_{\text{test}}$ defined as follows

$$A_{\mathcal{D}_{\text{test}}} = \frac{1}{|\mathcal{D}_{\text{test}}|} \sum_{i \in \mathcal{D}_{\text{test}}} A(\mathbf{a}_i), \quad (11)$$

and $\tilde{A}_{\mathcal{D}_{\text{test}}}$ denotes the accuracy computed analogously on steered outputs $\tilde{\mathbf{a}}_i$. SPI provides a bounded, symmetric

measure in $[-1, 1]$. If a steering method recovers the full performance, *i.e.* matches perfect accuracy, SPI equals 1. If steering degrades task accuracy fully to zero, SPI is $-1$. If there is no effect from steering, SPI will be 0. By measuring *relative* effects in this way, benchmarks across tasks, and models with varying baselines become more meaningful.

## 6.2. Methods

We evaluate our steering proposal against several baselines.

- **No steering.** Without intervention.

- **BASE-x (prompt-based steering).** Appends suffix *"Think before you answer."* to the prompt (see §A.5), serving as a prompt-based baseline (Kojima et al., 2022).

- **BASE-$\mu_k$ (contrastive steering).** Following (Rimsky et al., 2024; Tan et al., 2024), a steering vector is defined as the contrastive mean $(\mu_+ - \mu_-)$ at the last token position in the prompt $(k = n)$, using the layer identified as most effective by a probe $\hat{p}$ for steering. Contrastive pairs are constructed with the top-$k$ highest-, and lowest-error training samples $(k \in \{50, 100, 200\})$.

- **BASE-$\hat{p}$, BASE-$\hat{p}_{\text{log}}$ (probe-based steering).** Uses weights of linear error estimator $\hat{p}$ or logistic probe $\hat{p}_{\text{log}}$ (*i.e.* trained with the LM's predicted labels $\hat{y}$ (Cheng et al., 2024; von Rütte et al., 2024)) as additive steering directions without optimisation.

To assess if **MERA** improves existing baselines, we include the following methods.

- **MERA.** Trains a regressor $\hat{p}$ to predict the LM's error, optimising both steering direction, and strength (see §3).

- **MERA-$\hat{p}_{\text{log}}$.** Replaces $\hat{p}$ with weights of a logistic regression probe $\hat{p}_{\text{log}}$ (see **BASE-$\hat{p}_{\text{log}}$**).

- **MERA-$\mu_{100}$.** Substitutes probe weights $w$ by contrastive means, with $k = 100$ (see **BASE-$\mu_1$**).

For comparability, all **MERA** steering methods intervene across all layers, and token positions (see §4). In §A.6.4, we included an ablation study for such choices.

## 6.3. Results

Figure 4 provides an aggregate overview of steering performance across models, and datasets, averaging both evaluation modes (see §2.1), and error percentiles. Higher SPI values indicate stronger error mitigation, while negative SPI

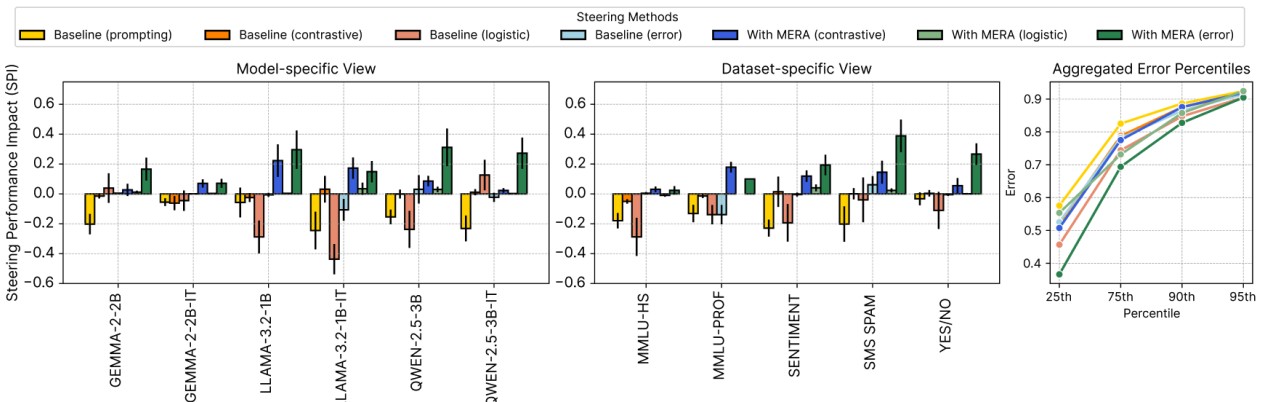

*Figure 4.* Overview of steering results for various LMs, and datasets. The first, and second panels display model-, and dataset-specific views, respectively, with $\delta = 0.05$. The third panel shows error percentiles, aggregated over the "last"n setting.

suggests degradation. In contrast, Table 1 offers a more detailed view, reporting SPI values, with $\delta = 0.01$ for each model, and dataset individually.

**MERA Outperforms Baselines, and Enhances Existing Steering Methods** The aggregate views in Figure 4, shows that **MERA** improves accuracy compared to both un-steered models, and existing steering approaches in all settings. The third panel in Figure 4 (right) shows that **MERA** reduces errors across the percentiles. While these overall positive trends are encouraging, Table 1 shows that performance gains can vary with the model, and dataset, as detailed next.

Certain LMs, and datasets benefit more than others. In Table 1, LLAMA-3.2-1B, and QWEN-2.5-3B show the largest improvements, particularly on binary tasks such as SMS SPAM, and YES/NO, where SPI gains reach +0.87. Base models benefit more from **MERA** steering compared to instruction-tuned models. Corroborating Rimsky et al. (2024), we note that the MMLU-HS, and MMLU-PROF subset is overall difficult to steer.

- Contrastive steering without **MERA** is highly unreliable, sometimes reducing accuracy (see Table 1 for examples). With **MERA**, it consistently improves accuracy, turning -0.05 SPI into +0.52 in YES/NO, and -0.09 into +0.21 in MMLU-HS, which is in line with the theoretical guarantees discussed in §3.2.

- Logistic probe-based steering mostly fails (-0.85 SPI in SENTIMENT) but with exceptions (+0.83 SPI SEN-TIMENT). **MERA** can help prevent task degradation (+0.00 SPI) where its base version fails.

- Our primary proposal **MERA** proves highly effective. Without **MERA**, **BASE**-$\hat{p}$ is too weak to be useful (+0.00

SPI in MMLU-HS, and SMS SPAM) with negligible or negative gains. With **MERA**, it becomes the best-performing method, improving SPI by +0.64 in SEN-TIMENT, and +0.53 in YES/NO, suggesting that error-based steering is better than using binary probe targets.

As detailed in §A.6.2, to analyse how steering affects prediction quality, we define transitions based on ground truth labels, where 0 denotes an incorrect, and 1 a correct prediction. Oversteering corresponds to unnecessary *degrading* outcomes ($1 \rightarrow 0$), while understeering reflects *missed opportunities* to correct ($0 \rightarrow 0$). As shown in Figure 12, while imperfect, **MERA** generally yields more *corrective* transitions with fewer degradations than baseline methods.

# 7. Discussion

Today, LMs can be frustratingly error-prone, failing even in simple supervised tasks. In this work, we establish a general-purpose methodology to steer LMs towards the correct answer. Unlike existing approaches that rely on expensive, uncalibrated, model-specific hyperparameter sweeps to find an appropriate strength for steering, our main contribution is a theoretically grounded framework that answers questions of *when, and how much* to steer.

Experiments across diverse datasets, and LM families consistently show that **MERA** not only improves task accuracy over baselines, but also enhances existing contrastive, and probe-based steering methods. However, our current framework is inherently linear, and thus limited in its capacity to capture non-linear relationships between internal activations, and model error. Our results on MMLU subsets underscore this limitation: certain datasets may be inherently harder to steer due to factors like semantic class overlap, task complexity, and label cardinality, warranting further investigation (Tan et al., 2024). One of **MERA**'s strengths

*Table 1.* Complete view of steering results for various LMs and tasks. SPI (Equation 10) is reported, capturing the relative increase or decrease in accuracy for both the "last" (left) and "exact" (right) evaluation modes for $\delta = 0.01$. Higher values are better.

| DATASET | METHOD | LLAMA-3.1-1B | LLAMA-3.1-1B-IT | GEMMA-2-2B | GEMMA-2-2B-IT | QWEN-2.5-3B | QWEN-2.5-3B-IT |
|---|---|---|---|---|---|---|---|
| YES/NO | **BASE**-x | (-0.05) \| (+0.10) | (+0.16) \| (+0.18) | (-0.21) \| (-0.39) | (-0.23) \| (+0.00) | (+0.00) \| (-0.01) | (+0.02) \| (+0.03) |
| | **BASE**-$\mu_{50}$ | (+0.00) \| (+0.03) | (-0.03) \| (+0.03) | (-0.01) \| (+0.00) | (-0.01) \| (-0.02) | (+0.02) \| (+0.00) | (+0.01) \| (+0.01) |
| | **BASE**-$\mu_{100}$ | (-0.05) \| (-0.06) | (+0.03) \| (+0.18) | (+0.01) \| (+0.00) | (-0.01) \| (-0.02) | (+0.01) \| (-0.03) | (-0.01) \| (+0.00) |
| | **BASE**-$\mu_{200}$ | (-0.04) \| (-0.06) | (-0.01) \| (+0.00) | (+0.01) \| (+0.01) | (+0.00) \| (+0.00) | (-0.01) \| (-0.01) | (+0.01) \| (+0.01) |
| | **BASE**-$\hat{p}_{\text{LOG}}$ | (+0.52) \| (-1.00) | (-0.28) \| (-1.00) | (-0.12) \| (+0.01) | (-0.02) \| (-0.11) | (+0.38) \| (+0.10) | (+0.49) \| (-0.31) |
| | **BASE**-$\hat{p}$ | (-0.01) \| (+0.01) | (-0.04) \| (-0.06) | (+0.00) \| (+0.00) | (+0.00) \| (+0.00) | (+0.01) \| (+0.01) | (+0.01) \| (+0.01) |
| | **MERA**-$\mu_{100}$ | (+0.52) \| (+0.00) | (+0.19) \| (+0.00) | (+0.00) \| (+0.00) | (+0.16) \| (+0.00) | (+0.00) \| (+0.00) | (+0.00) \| (+0.00) |
| | **MERA**-$\hat{p}_{\text{LOG}}$ | (+0.00) \| (+0.00) | (+0.00) \| (+0.00) | (+0.00) \| (+0.00) | (+0.00) \| (+0.00) | (+0.00) \| (+0.00) | (+0.00) \| (+0.00) |
| | **MERA** | (+0.52) \| (+0.28) | (+0.44) \| (+0.00) | (+0.42) \| (+0.00) | (+0.15) \| (+0.00) | (+0.47) \| (+0.52) | (+0.53) \| (+0.00) |
| SMS SPAM | **BASE**-x | (+0.79) \| (+0.03) | (-0.79) \| (-0.90) | (+0.00) \| (-1.00) | (-0.52) \| (+0.20) | (-0.69) \| (+0.44) | (+0.00) \| (+0.00) |
| | **BASE**-$\mu_{50}$ | (+0.84) \| (-1.00) | (-0.01) \| (-0.15) | (+0.00) \| (-0.10) | (+0.03) \| (+0.00) | (+0.15) \| (+0.00) | (+0.00) \| (+0.00) |
| | **BASE**-$\mu_{100}$ | (+0.06) \| (+0.10) | (-0.03) \| (-0.19) | (+0.00) \| (-0.19) | (+0.03) \| (+0.00) | (+0.24) \| (+0.00) | (+0.00) \| (+0.00) |
| | **BASE**-$\mu_{200}$ | (+0.00) \| (-0.05) | (+0.00) \| (+0.04) | (+0.00) \| (+0.00) | (+0.00) \| (+0.00) | (+0.20) \| (+0.04) | (+0.00) \| (+0.00) |
| | **BASE**-$\hat{p}_{\text{LOG}}$ | (+0.88) \| (-1.00) | (-0.03) \| (-1.00) | (+0.00) \| (+0.83) | (+0.19) \| (+0.00) | (+0.09) \| (-0.93) | (+0.48) \| (+0.02) |
| | **BASE**-$\hat{p}$ | (+0.01) \| (+0.02) | (+0.00) \| (+0.02) | (+0.00) \| (+0.00) | (-0.02) \| (+0.00) | (+0.70) \| (+0.00) | (+0.00) \| (+0.00) |
| | **MERA**-$\mu_{100}$ | (+0.44) \| (+0.77) | (+0.00) \| (+0.50) | (+0.00) \| (+0.00) | (+0.00) \| (+0.00) | (+0.00) \| (+0.00) | (+0.00) \| (+0.00) |
| | **MERA**-$\hat{p}_{\text{LOG}}$ | (+0.00) \| (+0.00) | (+0.00) \| (+0.00) | (+0.00) \| (+0.08) | (+0.00) \| (+0.00) | (+0.16) \| (+0.00) | (+0.00) \| (+0.00) |
| | **MERA** | (+0.87) \| (+0.71) | (+0.00) \| (+0.41) | (+0.00) \| (+0.12) | (+0.00) \| (+0.00) | (+0.83) \| (+0.70) | (+0.87) \| (+0.00) |
| SENTIMENT | **BASE**-x | (+0.00) \| (-0.65) | (-0.10) \| (+0.01) | (-0.32) \| (+0.09) | (+0.12) \| (-0.10) | (-0.45) \| (-0.35) | (-0.50) \| (-0.50) |
| | **BASE**-$\mu_{50}$ | (+0.00) \| (-0.33) | (+0.50) \| (+0.50) | (-0.19) \| (+0.00) | (-0.11) \| (-0.72) | (+0.03) \| (+0.01) | (+0.06) \| (+0.06) |
| | **BASE**-$\mu_{100}$ | (+0.00) \| (-0.28) | (+0.45) \| (+0.45) | (+0.00) \| (+0.00) | (-0.07) \| (-0.53) | (+0.02) \| (+0.00) | (+0.07) \| (+0.06) |
| | **BASE**-$\mu_{200}$ | (-0.04) \| (-0.06) | (-0.01) \| (-0.01) | (+0.01) \| (+0.01) | (+0.00) \| (+0.00) | (-0.01) \| (-0.01) | (+0.01) \| (+0.01) |
| | **BASE**-$\hat{p}_{\text{LOG}}$ | (-0.21) \| (-1.00) | (+0.87) \| (-1.00) | (-0.26) \| (-0.09) | (-0.05) \| (-0.54) | (-0.85) \| (+0.19) | (+0.38) \| (+0.24) |
| | **BASE**-$\hat{p}$ | (+0.00) \| (+0.03) | (-0.07) \| (-0.07) | (+0.00) \| (+0.00) | (+0.00) \| (+0.00) | (+0.06) \| (+0.00) | (+0.00) \| (+0.00) |
| | **MERA**-$\mu_{100}$ | (+0.00) \| (+0.00) | (+0.35) \| (+0.35) | (+0.00) \| (+0.00) | (+0.16) \| (+0.17) | (+0.24) \| (+0.11) | (+0.00) \| (+0.00) |
| | **MERA**-$\hat{p}_{\text{LOG}}$ | (+0.00) \| (+0.00) | (+0.21) \| (+0.21) | (+0.00) \| (+0.00) | (+0.00) \| (+0.00) | (+0.00) \| (+0.00) | (+0.00) \| (+0.00) |
| | **MERA** | (+0.00) \| (+0.00) | (+0.00) \| (+0.00) | (+0.49) \| (+0.33) | (+0.16) \| (+0.21) | (+0.00) \| (+0.00) | (+0.70) \| (+0.41) |
| MMLU-HS | **BASE**-x | (+0.16) \| (-0.59) | (-0.12) \| (-0.35) | (+0.00) \| (+0.00) | (+0.00) \| (+0.01) | (+0.00) \| (-0.26) | (-0.30) \| (-0.70) |
| | **BASE**-$\mu_{50}$ | (-0.02) \| (-0.03) | (-0.02) \| (+0.00) | (+0.00) \| (+0.00) | (+0.00) \| (+0.00) | (-0.05) \| (-0.08) | (-0.09) \| (-0.09) |
| | **BASE**-$\mu_{100}$ | (-0.05) \| (-0.05) | (-0.05) \| (-0.05) | (+0.00) \| (+0.00) | (+0.00) \| (-0.04) | (-0.09) \| (-0.12) | (-0.07) \| (-0.07) |
| | **BASE**-$\mu_{200}$ | (+0.01) \| (-0.03) | (+0.00) \| (-0.05) | (+0.00) \| (+0.00) | (+0.00) \| (+0.01) | (+0.05) \| (-0.08) | (-0.01) \| (-0.01) |
| | **BASE**-$\hat{p}_{\text{LOG}}$ | (+0.00) \| (+0.00) | (+0.00) \| (+0.00) | (+0.00) \| (+0.00) | (+0.00) \| (+0.00) | (+0.00) \| (+0.00) | (+0.00) \| (+0.00) |
| | **BASE**-$\hat{p}$ | (+0.00) \| (+0.00) | (+0.01) \| (-0.07) | (+0.00) \| (+0.05) | (+0.00) \| (+0.01) | (+0.02) \| (+0.00) | (+0.02) \| (+0.02) |
| | **MERA**-$\mu_{100}$ | (+0.00) \| (+0.00) | (+0.00) \| (+0.00) | (+0.00) \| (+0.00) | (+0.00) \| (+0.00) | (+0.21) \| (+0.13) | (+0.00) \| (+0.00) |
| | **MERA**-$\hat{p}_{\text{LOG}}$ | (+0.00) \| (+0.00) | (+0.00) \| (+0.00) | (+0.00) \| (+0.00) | (+0.00) \| (+0.00) | (+0.21) \| (+0.13) | (+0.00) \| (+0.00) |
| | **MERA** | (+0.00) \| (+0.00) | (+0.33) \| (+0.00) | (+0.00) \| (+0.00) | (+0.00) \| (+0.00) | (+0.00) \| (+0.00) | (-0.01) \| (-0.01) |
| MMLU-PROF | **BASE**-x | (+0.01) \| (-0.63) | (-0.16) \| (-0.42) | (+0.00) \| (+0.01) | (+0.03) \| (+0.01) | (+0.01) \| (-0.15) | (+0.01) \| (-0.27) |
| | **BASE**-$\mu_{50}$ | (-0.06) \| (-0.06) | (-0.19) \| (-0.07) | (+0.00) \| (+0.00) | (+0.00) \| (-0.12) | (-0.03) \| (-0.02) | (+0.07) \| (+0.07) |
| | **BASE**-$\mu_{100}$ | (+0.01) \| (+0.01) | (-0.11) \| (-0.09) | (+0.00) \| (+0.00) | (+0.00) \| (-0.05) | (-0.03) \| (+0.01) | (+0.04) \| (+0.04) |
| | **BASE**-$\mu_{200}$ | (+0.02) \| (+0.01) | (-0.05) \| (-0.13) | (+0.00) \| (+0.00) | (+0.00) \| (-0.05) | (-0.04) \| (-0.02) | (+0.01) \| (+0.01) |
| | **BASE**-$\hat{p}_{\text{LOG}}$ | (-0.06) \| (-0.06) | (+0.01) \| (-0.80) | (+0.00) \| (+0.00) | (+0.01) \| (+0.02) | (-0.24) \| (-0.25) | (+0.06) \| (-0.35) |
| | **BASE**-$\hat{p}$ | (-0.06) \| (-0.06) | (+0.01) \| (-0.80) | (+0.00) \| (+0.00) | (+0.01) \| (+0.02) | (-0.24) \| (-0.25) | (+0.06) \| (-0.35) |
| | **MERA**-$\mu_{100}$ | (+0.00) \| (+0.00) | (+0.00) \| (+0.00) | (+0.00) \| (+0.24) | (+0.00) \| (+0.00) | (+0.00) \| (+0.00) | (+0.00) \| (+0.00) |
| | **MERA**-$\hat{p}_{\text{LOG}}$ | (+0.00) \| (+0.00) | (+0.00) \| (+0.00) | (+0.00) \| (+0.00) | (+0.00) \| (+0.00) | (+0.00) \| (+0.00) | (+0.00) \| (+0.00) |
| | **MERA** | (+0.00) \| (+0.00) | (+0.00) \| (+0.00) | (+0.00) \| (+0.00) | (+0.00) \| (+0.00) | (+0.00) \| (+0.00) | (+0.10) \| (+0.10) |

is to detect such *unsteerable* cases, and abstain from intervention, making it a more cautious steering approach compared to baseline methods. That said, while **MERA** improves targeted error correction, it may inadvertently impair general capabilities. For example, steering for specalised tasks could reduce performance on unrelated generation or reasoning tasks. Exploring these trade-offs, and developing safeguards to preserve generality, and text fluency is an important direction for future work.

In this work, we focus our evaluation on MCQA classification tasks, as it allows for a *controlled* setup where ground-truth error can be unambiguously read off from token-level predictions (see §2). However, this choice of setting does not reflect a fundamental limitation of **MERA**. The framework is applicable to any task where labeled supervision over model behavior is available. For example, **MERA** could be extended to alignment objectives such as toxicity reduction, refusal behavior, and harmfulness mitigation; directions that we plan to explore next.

There are also several promising directions for advancing **MERA** as a general steering methodology. One promising avenue is to replace the linear error probes with non-linear estimators, such as neural networks, an extension we outline in §A.1. Another is investigating the impact of the evaluation metric used during calibration. For example, could using F1 score instead of accuracy promote more balanced steering in tasks with high class imbalance or cardinality? We aim to explore such questions in future work.

## Acknowledgements

We thank Ronald Namwanza for going above, and beyond to ensure we had access to computing resources during critical phases of this research.

## Disclaimer

## Impact Statement

This paper presents a method for steering language models to reduce their error on well-defined prediction tasks. Given the increasing appetite for applying such models on real-world problems, the responsible deployment of more mature versions of this technology could have a positive societal impact, enabling more aligned, and trustworthy AI products, and services.

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

# A. Appendix

## A.1. Non-Linear Case

When $\hat{p}$ is replaced by a non-linear function (*e.g.* a MLP), the constraint $\hat{p}(h + v) \leq \alpha$ cannot be expressed in a simple linear form. In such cases, a closed-form solution does not exist.

One approach is to apply a *first-order Taylor approximation* of the non-linear function $\hat{p}(h + v)$ around the current representation $h$. Specifically, we linearise $\hat{p}$ as

$$\hat{p}(h + v) \approx \hat{p}(h) + \nabla_h \hat{p}(h)^\top v,$$

where $\nabla_h \hat{p}(h) \in \mathbb{R}^d$ is the gradient of $\hat{p}$ with respect to $h$. Substituting this into the constraint $\hat{p}(h + v) \leq \alpha$ yields the approximate linear constraint:

$$\hat{p}(h) + \nabla_h \hat{p}(h)^\top v \leq \alpha.$$

This transforms the problem into the following *quadratic program*

$$\min_v \|v\|_2^2 \quad \text{s.t.} \quad \nabla_h \hat{p}(h)^\top v \leq \alpha - \hat{p}(h).$$

which corresponds exactly to the problem solved in the main body in Section 3.1 by letting $w := \nabla_h \hat{p}(h)$.

Alternatively, we can directly solve the non-linear constrained problem using numerical optimization techniques, such as projected gradient methods or interior-point solvers or adopting a *penalty-based* approach that converts the constraint into a differentiable penalty term, resulting in the following unconstrained objective

$$\min_v \left\{ \|v\|_2^2 + \zeta \cdot \left( \max(0, \hat{p}(h + v) - \alpha) \right)^2 \right\}, \tag{12}$$

where $\zeta > 0$ is a scalar hyperparameter controlling how strictly we enforce the constraint. Because this formulation is fully differentiable, any standard gradient-based optimiser (*e.g.* Adam or SGD) can be used to iteratively update $v$ until convergence.

Similarly to the linear case, we choose $\alpha$, and $\delta$ using a calibration set to effectively minimise the true loss. However, note that each candidate value requires solving the above problem 12 anew, which can be computationally demanding.

## A.2. Calibration Step Detailed, and Theoretical Guarantees

The rationale behind our selection procedure is to ensure that we do not choose a value of $\alpha \in (0, 1)$ unless it demonstrates a statistically significant improvement in performance. The strategy is as follows: we first identify the subset of $\alpha$ values that provably yield a positive performance improvement with high probability, and then select the one among them with the highest empirical performance.

To formalise this, suppose $f(\alpha, X_i) \in [0, 1]$ is a performance function, and $X_i$ is a random input variable. Then, the empirical performance on a calibration dataset $D_n = \{X_1, \ldots, X_n\}$ is given by

$$P(\alpha, D_n) = \frac{1}{n} \sum_{i=1}^n f(\alpha, X_i).$$

We denote $P(\alpha, D)$ as the theoretical performance.

Our approach proceeds in the following steps:

1. **Discretise** the interval $[0, 1]$ into $K$ points: $\{\alpha_1, \ldots, \alpha_K\}$.

2. **Construct confidence bands** around each $\alpha_j$, using Hoeffding's inequality, for example

$$\Pr\left( |P(\alpha_j, D_n) - P(\alpha_j, D)| \leq \delta_n \right) \geq 1 - \frac{\alpha}{K},$$

where $\delta_n = \sqrt{\frac{\ln(2K/\alpha)}{2n}}$.

3. **Apply the union bound** to ensure that, with probability at least $1 - \alpha$, all the above intervals simultaneously hold

$$|P(\alpha_j, D_n) - P(\alpha_j, D)| \leq \delta_n \quad \text{for all } j = \{1, \ldots, K\},$$

or equivalently

$$\Pr\left(\sup_{\alpha \in [0,1]} |P(\alpha, D_n) - P(\alpha, D)| \leq \varepsilon_n\right) \geq 1 - \alpha.$$

With this in place, we define the valid set

$$\alpha_{\text{valid}} = \{\alpha : L(\alpha) > 0\},$$

and select

$$\alpha^* = \arg\max_{\alpha \in \alpha_{\text{valid}}} P(\alpha, D_n).$$

Because the confidence bands hold uniformly over all $\alpha$ with high probability, we can guarantee that for every $\alpha \in \alpha_{\text{valid}}$, the true performance $P(\alpha, D)$ is also positive. Thus, on this "good event", $\alpha^*$ indeed corresponds to a true positive performance improvement.

This provides a formal guarantee, based on i.i.d. data, and bounded performance function, that our method yields a statistically sound improvement or abstains with high probability.

**Alternative Perspective**    An alternative to Bonferroni-style correction, which may sometimes be conservative is to **split the data** into two parts:

- Use one part to select the empirically optimal $\alpha^\star$,

- Then use the held-out part to estimate a confidence interval for the performance metric.

We accept the selected $\alpha^\star$ only if the lower bound of its confidence interval exceeds the desired threshold. This approach avoids the potentially conservative bounds introduced by Bonferroni correction, and can lead to tighter, and more adaptive inference.

### A.3. Related Works on Steering Strength

Several prior works have explored different strategies for determining the appropriate strength of steering in LMs. We discuss these below.

**Fixed Strengths, and Hyperparameter Sweeps**    Li et al. (2023); Liu et al. (2024); Turner et al. (2024); von Rütte et al. (2024); Postmus & Abreu (2024); Rimsky et al. (2024); Tan et al. (2024); Cao et al. (2024) rely on brute-force testing of different steering strengths across layers, and token positions, selecting the best-performing configuration. For example, Postmus & Abreu (2024); Turner et al. (2024); Liu et al. (2024); Cao et al. (2024) performs an exhaustive grid search over multiple layers, and multipliers, determining steering strength via empirical accuracy. Similarly, von Rütte et al. (2024) evaluates steering at fixed increments for LLaMA, and MISTRAL, while adjusting layers. Durmus et al. (2024) uses a heuristic range of (-5,5) across tasks, admittedly an "arbitrary" decision. This approach is computationally expensive, and not generalisable across architectures or tasks.

Arditi et al. (2024); Bhattacharjee et al. (2024); Ball et al. (2024) bypass search entirely, applying a pre-set steering strength across all instances, fixing $\lambda$ at 1. While computationally efficient, unlike hyperparameter sweeps, no effort is made to select an optimal $\lambda^\star$.

**Conditional Methods**    Wang et al. (2025); Scalena et al. (2024) regulate steering strength dynamically. For example, Wang et al. (2025) clusters attention heads based on truthfulness, applying a variable steering strength that scales inversely with classifier confidence. Scalena et al. (2024) diminishes steering strength throughout model generation, reducing intervention as logit divergence stabilises. Cheng et al. (2024) is most similar to our work, and formulates steering as a constrained minimisation problem, selecting the minimal $\lambda$ necessary to satisfy a safety threshold. A limitation of these methods is that they rely on indirect heuristics rather than directly *calibrating* for error minimisation on a given task.

Lee et al. (2025) also explore conditional steering, but differ from our approach in that they trigger interventions based on cosine similarity with condition vectors, use PCA on contrastive examples to define steering directions, and select intervention strength via grid search rather than a closed-form solution. Also, Luo et al. (2024) propose an intervention method with some mathematical resemblance to ours, modulating steering strength via inner products, but differ in motivation, and design, as their approach targets suppression of harmful concepts in a predefined dictionary via supervised training, whereas MERA focuses on error mitigation with calibrated, layer-wise closed-form updates derived from held-out data.

### A.4. Related Works on Steering Options

To extract activations, contrastive steering often relies on a specific model layer (Arditi et al., 2024; Rimsky et al., 2024; Turner et al., 2024), and the last token in the input prompt (Rimsky et al., 2024; Wang et al., 2025; Scalena et al., 2024), *i.e.* $k = n$. The rationale behind using the last token position is that, with well-constructed prompts, and well-defined tasks such as MCQA, the functional behaviour of the model can be captured locally at a single token position (Arditi et al., 2024; Rimsky et al., 2024). Steering is then *applied* variously. That is, across all layers (Cheng et al., 2024; Liu et al., 2024), on specific layers (Arditi et al., 2024; Rimsky et al., 2024; Marks & Tegmark, 2024), targeting all token positions (Cheng et al., 2024) or selected ones, such as post-instruction or generated tokens (Arditi et al., 2024; Rimsky et al., 2024).

The generalisability of these extraction, and application strategies across tasks, and models remains an open question. We contribute by addressing it in §4.

### A.5. Experimental Details

Tables 2, and 3 summarise the datasets, and models used in our experiments. Table 2 provides an overview of the datasets, including the number of classes, class distributions, and sample counts across training, calibration, and test sets. Table 3 details the models, specifying the number of layers, type, and parameter size.

*Table 2.* Datasets overview.

| DATASET | NR. CLASSES | CLASS LABELS (DIST.) | NR. TRAIN./CAL./TEST |
|---|---|---|---|
| YES/NO (AI, 2023) | 2 | YES (32.5), NO (67.5) | 3000 / 250 / 250 |
| SMS SPAM (ALMEIDA ET AL., 2011) | 2 | SPAM (86.4), HAM (13.6) | 3000 / 250 / 250 |
| SENTIMENT (AI, 2023) | 3 | POS (41.7), NEG (55.3), NEU (3.0) | 3000 / 250 / 250 |
| MMLU-HS (HENDRYCKS ET AL., 2021A) | 4 | A (20.7), B (23.9), C (24.8), D (30.7) | 3000 / 210 / 210 |
| MMLU-PROF (HENDRYCKS ET AL., 2021A) | 4 | A, B, C, D | 2601 / 210 / 210 |

*Table 3.* Models overview.

| MODEL | NR. LAYERS | TYPE | PARAMETERS |
|---|---|---|---|
| LLAMA-3.1-1B (TEAM, 2024B) | 16 | IT | 1B |
| LLAMA-3.1-1B-IT (TEAM, 2024B) | 16 | BASE | 1B |
| GEMMA-2-2B (TEAM, 2024A) | 26 | BASE | 2B |
| GEMMA-2-2B-IT (TEAM, 2024A) | 26 | IT | 2B |
| QWEN-2.5-3B (TEAM, 2024C) | 36 | BASE | 3B |
| QWEN-2.5-3B-IT (TEAM, 2024C) | 36 | IT | 3B |

**Unsteered Task Accuracy**  Table 4 provides an overview of unsteered task accuracy across different models, presenting accuracy on the test set (left), and error (right) for various tasks in both evaluation modes.

**Absolute Differences in Task Accuracy**  Table 5 provides an overview of the absolute difference between unsteered, and unsteered task accuracy across different models, presenting raw accuracy deltas in the *"last"* mode (left), and in the *"exact"* mode (right) for various tasks, and each steering method.

**Hardware**  All experiments were conducted on AWS instances, primarily using the G5.16XLARGE instances with NVIDIA A10G GPUs. SAE caching experiments were exclusively run on G5.48XLARGE instances, offering enhanced 8 GPUs capacity.

*Table 4.* **Unsteered task accuracy** overview. Accuracy on the test dataset (left), and error (right) across datasets in both evaluation modes.

| MODEL | MODE | YES/NO | SMS SPAM | SENTIMENT | MMLU-HS | MMLU-PROF |
|---|---|---|---|---|---|---|
| LLAMA-3.2-1B (BASE) | *"last"* | (0.336 \| 0.571) | (0.128 \| 0.622) | (0.056 \| 0.893) | (0.190 \| 0.776) | (0.267 \| 0.561) |
| | *"exact"* | (0.380 \| 0.311) | (0.172 \| 0.590) | (0.376 \| 0.467) | (0.176 \| 0.717) | (0.409 \| 0.543) |
| LLAMA-3.2-1B (IT) | *"last"* | (0.436 \| 0.519) | (0.892 \| 0.179) | (0.236 \| 0.725) | (0.195 \| 0.779) | (0.441 \| 0.559) |
| | *"exact"* | (0.428 \| 0.471) | (0.784 \| 0.201) | (0.236 \| 0.725) | (0.205 \| 0.721) | (0.441 \| 0.559) |
| GEMMA-2B (BASE) | *"last"* | (0.452 \| 0.519) | (0.108 \| 0.741) | (0.124 \| 0.784) | (0.152 \| 0.814) | (0.262 \| 0.203) |
| | *"exact"* | (0.384 \| 0.510) | (0.084 \| 0.146) | (0.828 \| 0.320) | (0.000 \| 0.000) | (0.262 \| 0.203) |
| GEMMA-2B (IT) | *"last"* | (0.644 \| 0.394) | (0.224 \| 0.699) | (0.900 \| 0.135) | (0.171 \| 0.800) | (0.437 \| 0.552) |
| | *"exact"* | (0.428 \| 0.456) | (0.108 \| 0.869) | (0.884 \| 0.335) | (0.110 \| 0.512) | (0.339 \| 0.555) |
| QWEN-2.5-3B (BASE) | *"last"* | (0.356 \| 0.575) | (0.356 \| 0.535) | (0.364 \| 0.714) | (0.267 \| 0.719) | (0.332 \| 0.660) |
| | *"exact"* | (0.360 \| 0.571) | (0.108 \| 0.831) | (0.092 \| 0.303) | (0.238 \| 0.633) | (0.220 \| 0.589) |
| QWEN-2.5-3B (IT) | *"last"* | (0.328 \| 0.642) | (0.108 \| 0.892) | (0.072 \| 0.690) | (0.319 \| 0.709) | (0.204 \| 0.742) |
| | *"exact"* | (0.324 \| 0.647) | (0.108 \| 0.892) | (0.072 \| 0.690) | (0.319 \| 0.709) | (0.203 \| 0.743) |

*Table 5.* Complete view of absolute difference steering results for various LMs and tasks. Absolute difference is reported, capturing the increase or decrease in accuracy for both the "last" (left) and "exact" (right) evaluation modes for $\delta = 0.01$. Higher values are better.

| DATASET | METHOD | LLAMA-3.1-1B | LLAMA-3.1-1B-IT | GEMMA-2-2B | GEMMA-2-2B-IT | QWEN-2.5-3B | QWEN-2.5-3B-IT |
|---|---|---|---|---|---|---|---|
| YES/NO | **BASE**-x | (-0.02) \| (+0.06) | (+0.09) \| (+0.10) | (-0.10) \| (-0.15) | (-0.15) \| (+0.00) | (+0.00) \| (+0.00) | (+0.02) \| (+0.02) |
| | **BASE**-$\mu_{50}$ | (+0.00) \| (+0.02) | (-0.01) \| (+0.02) | (+0.00) \| (+0.00) | (-0.01) \| (-0.01) | (+0.02) \| (+0.00) | (+0.00) \| (+0.00) |
| | **BASE**-$\mu_{100}$ | (-0.02) \| (-0.02) | (+0.02) \| (+0.10) | (+0.01) \| (+0.00) | (+0.00) \| (-0.01) | (+0.00) \| (-0.01) | (+0.00) \| (+0.00) |
| | **BASE**-$\mu_{200}$ | (+0.12) \| (-0.24) | (-0.07) \| (+0.03) | (-0.04) \| (-0.21) | (-0.18) \| (+0.06) | (+0.00) \| (+0.00) | (+0.29) \| (+0.30) |
| | **BASE**-$\hat{p}_{\text{LOG}}$ | (+0.35) \| (-0.38) | (-0.12) \| (-0.43) | (-0.05) \| (+0.01) | (-0.01) \| (-0.05) | (+0.24) \| (+0.06) | (+0.33) \| (-0.10) |
| | **BASE**-$\hat{p}$ | (+0.00) \| (+0.00) | (-0.02) \| (-0.02) | (+0.00) \| (+0.00) | (+0.00) \| (+0.00) | (+0.01) \| (+0.00) | (+0.01) \| (+0.01) |
| | **MERA**-$\mu_{100}$ | (+0.35) \| (+0.00) | (+0.11) \| (+0.00) | (+0.00) \| (-0.08) | (+0.06) \| (+0.00) | (+0.00) \| (+0.00) | (+0.00) \| (+0.00) |
| | **MERA**-$\hat{p}_{\text{LOG}}$ | (+0.00) \| (+0.00) | (-0.01) \| (-0.02) | (+0.00) \| (+0.02) | (+0.00) \| (+0.01) | (+0.01) \| (+0.01) | (+0.00) \| (+0.01) |
| | **MERA** | (+0.35) \| (+0.18) | (+0.25) \| (+0.00) | (+0.23) \| (-0.04) | (+0.05) \| (+0.00) | (+0.30) \| (+0.34) | (+0.36) \| (-0.02) |
| SMS SPAM | **BASE**-x | (+0.69) \| (+0.03) | (-0.70) \| (-0.71) | (+0.00) \| (-0.08) | (-0.12) \| (+0.18) | (-0.24) \| (+0.40) | (+0.00) \| (+0.00) |
| | **BASE**-$\mu_{50}$ | (+0.74) \| (-0.17) | (-0.01) \| (-0.12) | (+0.00) \| (-0.01) | (+0.02) \| (+0.00) | (+0.10) \| (+0.00) | (+0.00) \| (+0.00) |
| | **BASE**-$\mu_{100}$ | (+0.05) \| (+0.08) | (-0.02) \| (-0.15) | (+0.00) \| (-0.02) | (+0.02) \| (+0.00) | (+0.16) \| (+0.00) | (+0.00) \| (+0.00) |
| | **BASE**-$\mu_{200}$ | (+0.04) \| (-1.00) | (+0.02) \| (+0.00) | (-0.09) \| (-0.14) | (-0.02) \| (+0.06) | (+0.00) \| (+0.00) | (+0.00) \| (+0.00) |
| | **BASE**-$\hat{p}_{\text{LOG}}$ | (+0.76) \| (-0.17) | (-0.02) \| (-0.78) | (+0.00) \| (+0.76) | (+0.15) \| (+0.00) | (+0.06) \| (-0.10) | (+0.43) \| (+0.02) |
| | **BASE**-$\hat{p}$ | (+0.01) \| (+0.02) | (+0.00) \| (+0.00) | (+0.00) \| (+0.00) | (+0.00) \| (+0.00) | (+0.45) \| (+0.00) | (+0.00) \| (+0.00) |
| | **MERA**-$\mu_{100}$ | (+0.38) \| (+0.64) | (+0.00) \| (+0.11) | (+0.00) \| (+0.00) | (+0.03) \| (+0.00) | (+0.00) \| (+0.00) | (+0.00) \| (+0.00) |
| | **MERA**-$\hat{p}_{\text{LOG}}$ | (+0.02) \| (+0.00) | (+0.00) \| (+0.00) | (+0.00) \| (+0.07) | (+0.00) \| (+0.00) | (+0.10) \| (+0.02) | (+0.00) \| (+0.00) |
| | **MERA** | (+0.76) \| (+0.59) | (+0.00) \| (+0.09) | (+0.06) \| (+0.11) | (+0.00) \| (+0.00) | (+0.53) \| (+0.62) | (+0.78) \| (+0.08) |
| SENTIMENT | **BASE**-x | (+0.00) \| (-0.24) | (-0.02) \| (+0.01) | (-0.04) \| (+0.02) | (+0.01) \| (-0.09) | (-0.16) \| (-0.03) | (-0.04) \| (-0.04) |
| | **BASE**-$\mu_{50}$ | (+0.00) \| (-0.12) | (+0.38) \| (+0.38) | (-0.02) \| (+0.00) | (-0.10) \| (-0.64) | (+0.02) \| (+0.01) | (+0.06) \| (+0.06) |
| | **BASE**-$\mu_{100}$ | (+0.00) \| (-0.10) | (+0.34) \| (+0.34) | (+0.00) \| (+0.00) | (-0.06) \| (-0.47) | (+0.01) \| (+0.00) | (+0.06) \| (+0.06) |
| | **BASE**-$\mu_{200}$ | (-0.35) \| (+0.35) | (-0.44) \| (-0.92) | (+0.00) \| (-0.61) | (+0.23) \| (+0.36) | (-0.79) \| (-0.96) | (+0.39) \| (+0.10) |
| | **BASE**-$\hat{p}_{\text{LOG}}$ | (-0.01) \| (-0.38) | (+0.67) \| (-0.24) | (-0.03) \| (-0.08) | (-0.05) \| (-0.48) | (-0.31) \| (+0.17) | (+0.36) \| (+0.22) |
| | **BASE**-$\hat{p}$ | (+0.00) \| (+0.02) | (-0.02) \| (-0.02) | (+0.00) \| (+0.00) | (+0.00) \| (+0.00) | (+0.04) \| (+0.00) | (+0.00) \| (+0.00) |
| | **MERA**-$\mu_{100}$ | (+0.00) \| (+0.03) | (+0.26) \| (+0.26) | (+0.00) \| (+0.00) | (+0.02) \| (+0.02) | (+0.15) \| (+0.10) | (+0.00) \| (+0.00) |
| | **MERA**-$\hat{p}_{\text{LOG}}$ | (+0.00) \| (+0.01) | (+0.16) \| (+0.16) | (+0.00) \| (-0.01) | (+0.00) \| (+0.00) | (+0.02) \| (+0.01) | (+0.00) \| (+0.00) |
| | **MERA** | (+0.00) \| (+0.00) | (+0.00) \| (+0.00) | (+0.43) \| (+0.06) | (+0.02) \| (+0.02) | (+0.00) \| (+0.03) | (+0.65) \| (+0.38) |
| MMLU-HS | **BASE**-x | +0.13 \| (-0.10) | (-0.02) \| (-0.07) | (+0.00) \| (+0.00) | (+0.00) \| (+0.01) | (+0.00) \| (-0.06) | (-0.10) \| (-0.22) |
| | **BASE**-$\mu_{50}$ | (+0.00) \| (+0.00) | (+0.00) \| (+0.00) | (+0.00) \| (+0.00) | (+0.00) \| (+0.00) | (-0.01) \| (-0.02) | (-0.03) \| (-0.03) |
| | **BASE**-$\mu_{100}$ | (-0.01) \| (-0.01) | (-0.01) \| (-0.01) | (+0.00) \| (+0.00) | (+0.00) \| (+0.00) | (-0.02) \| (-0.03) | (-0.02) \| (-0.02) |
| | **BASE**-$\mu_{200}$ | (+0.00) \| (+0.00) | (+0.04) \| (+0.06) | (-0.10) \| (-0.49) | (+0.03) \| (+0.02) | (+0.03) \| (-0.32) | (-0.21) \| (-0.24) |
| | **BASE**-$\hat{p}_{\text{LOG}}$ | (+0.00) \| (+0.00) | (+0.00) \| (+0.00) | (+0.00) \| (+0.00) | (+0.00) \| (+0.00) | (+0.00) \| (+0.00) | (+0.00) \| (+0.00) |
| | **BASE**-$\hat{p}$ | (+0.00) \| (+0.00) | (+0.01) \| (-0.01) | (+0.00) \| (+0.05) | (+0.00) \| (+0.00) | (+0.01) \| (+0.00) | (+0.01) \| (+0.01) |
| | **MERA**-$\mu_{100}$ | (+0.00) \| (+0.00) | (+0.00) \| (+0.00) | (+0.00) \| (+0.00) | (+0.00) \| (+0.03) | (+0.15) \| (+0.10) | (+0.00) \| (+0.00) |
| | **MERA**-$\hat{p}_{\text{LOG}}$ | (+0.00) \| (+0.00) | (+0.00) \| (+0.00) | (+0.00) \| (+0.00) | (+0.00) \| (+0.00) | (+0.14) \| (+0.10) | (+0.00) \| (+0.00) |
| | **MERA** | (+0.00) \| (+0.00) | (+0.26) \| (+0.01) | (+0.00) \| (+0.00) | (+0.03) \| (+0.01) | (-0.01) \| (+0.00) | (+0.00) \| (+0.00) |
| MMLU-PROF | **BASE**-x | (+0.00) \| (-0.15) | (-0.05) \| (-0.11) | (+0.00) \| (+0.01) | (+0.02) \| (+0.00) | (+0.00) \| (-0.05) | (+0.00) \| (-0.05) |
| | **BASE**-$\mu_{50}$ | (-0.01) \| (-0.01) | (-0.06) \| (-0.02) | (+0.00) \| (+0.00) | (+0.00) \| (-0.02) | (-0.01) \| (+0.00) | (+0.06) \| (+0.06) |
| | **BASE**-$\mu_{100}$ | (+0.01) \| (+0.01) | (-0.03) \| (-0.02) | (+0.00) \| (+0.00) | (+0.00) \| (-0.01) | (-0.01) \| (+0.00) | (+0.03) \| (+0.03) |
| | **BASE**-$\mu_{200}$ | (+0.01) \| (+0.01) | (-0.01) \| (-0.03) | (+0.00) \| (+0.00) | (+0.00) \| (-0.01) | (-0.01) \| (+0.00) | (+0.01) \| (+0.01) |
| | **BASE**-$\hat{p}_{\text{LOG}}$ | (-0.01) \| (-0.01) | (+0.00) \| (-0.21) | (+0.00) \| (+0.00) | (+0.00) \| (+0.01) | (-0.08) \| (-0.08) | (+0.05) \| (-0.07) |
| | **BASE**-$\hat{p}$ | (-0.01) \| (-0.01) | (+0.00) \| (-0.21) | (+0.00) \| (+0.00) | (+0.00) \| (+0.01) | (-0.08) \| (-0.08) | (+0.05) \| (-0.07) |
| | **MERA**-$\mu_{100}$ | (+0.00) \| (+0.00) | (+0.00) \| (+0.00) | (+0.00) \| (+0.24) | (+0.00) \| (+0.00) | (+0.00) \| (+0.00) | (+0.00) \| (+0.00) |
| | **MERA**-$\hat{p}_{\text{LOG}}$ | (+0.00) \| (+0.00) | (+0.00) \| (+0.00) | (+0.00) \| (+0.00) | (+0.00) \| (+0.00) | (+0.00) \| (+0.00) | (+0.00) \| (+0.00) |
| | **MERA** | (+0.00) \| (+0.00) | (+0.00) \| (+0.00) | (+0.00) \| (+0.00) | (+0.00) \| (+0.00) | (+0.00) \| (+0.00) | (+0.08) \| (+0.08) |

**Runtime**    MERA introduces two kinds of overhead: one at deployment, and one during calibration. At deployment, MERA incurs negligible runtime cost, as the learned steering vector $v$ can be applied via a standard *forward hook*. The cost is equivalent to a single vector addition per intervention layer, and token position.

The main computational cost arises during calibration, after the probes have been trained, where a held-out calibration dataset is used to select the optimal confidence threshold $\alpha$. This process involves evaluating a small number of candidate $\alpha$ values (e.g., 10-20) across a modest calibration set (e.g., 250 examples), computing the closed-form solution for each. Since this is a one-time cost per model, and dataset pair, it is amortised across deployment.

In practice, calibration takes between 10 minutes and 1 hour on our hardware (see §A.5), though actual runtime depends on model size, input length, and hardware configuration.

**Constructing Prompts**    Prompts are tailored to each dataset. Each dataset contains 3000 training samples, 250 calibration samples, and 250 test samples (except MMLU-HS, and MMLU-PROF, with 2061 training samples, and 210 calibration/test samples available). Below are the prompt templates for the datasets used in the experiments.

```
Question:  [question text]{Options: A. [Option A] B. [Option B] C. [Option C] D.
[Option D] Please select the correct answer.  Only return one letter:  A, B, C, or D.
Answer:\n
```

*Figure 5.* Prompt template for MMLU subsets.

```
This SMS (text message):  "[SMS text]" is classified as either spam or ham. Please
evaluate the content of the SMS, and select the correct classification. Only return one
word:  "ham" or "spam". Answer:\n
```

*Figure 6.* Prompt template for SMS SPAM.

```
You are a financial sentiment analysis expert.  Your task is to analyze the sentiment
expressed in the given financial text.  Only reply with positive, neutral, or negative.
Financial text:  [financial text] Answer:\n
```

*Figure 7.* Prompt template for SENTIMENT.

```
You are a financial expert.  Your task is to answer yes/no questions based on the given
headline or news content.  Context:  Headline:  "[headline text]" Now answer this
question:  [yes/no question] Answer:\n
```

*Figure 8.* Prompt template for YES/NO.

**Parsing Model Completions**    To compute the error $E(\mathbf{a})$, and labels $\hat{y}$, we extract the exact answers directly from model completions, bypassing the need for external sources such as additional LMs to assess the ground truth (Orgad et al., 2024; Arditi et al., 2024). This approach focuses on tasks where the correct label can be reliably identified at a single token position, simplifying the evaluation process while ensuring a systematic, and linguistically informed parsing.

For each task, class labels are transformed into multiple semantic variants, including lowercase, uppercase, and capitalised forms, with optional whitespace or newline prefixes. These string variants are tokenised, and their final token IDs are extracted for each ground truth label, and the LM's vocabulary.

During evaluation, we employ flexible matching within the token ID space, where any token ID corresponding to the semantic variants of a class is considered valid. The model's output, $\mathbf{a}$, is scanned starting from a specified position, and the first occurrence of a matching token ID is recorded. If no match is found, a fallback index of $-1$ is assigned, indicating no valid prediction. In the *"exact"* evaluation mode, a prediction is considered correct if, and only if the predicted token matches the class token IDs.

**Training Linear Probes**    For each layer of a given LM, we train a distinct set of linear probes given the training samples, which are split into 70% for, and 30% for validation. The input features, $\mathbf{X} \in \mathbb{R}^{N \times d}$, were extracted from residual stream activations at specific token positions (exact answer from open-ended model completion or the last prompt position) across all layers. The target variable is the error $E(\mathbf{a})$ of the LM.

To account for random variability, for each setting (*i.e.* layer, and token position), we trained 5 linear regression models using Lasso regularisation strengths $\eta \in \{0.005, 0.01, 0.05, 0.1, 0.25, 0.5\}$, alongside an unregularised linear regression with no sparsity. For simplicity, we omit the bias term in linear models, and focus on the weight vector. For each layer, we selected the model weights with the lowest RMSE on the validation set.

For the logistic probes, we follow a similar methodology but the target variable is the accuracy $A(\mathbf{a})$.

## A.6. Extended Results

### A.6.1. PROBE PERFORMANCE

Figures 9-10 show an extended version of Figure 2, separated by distinct datasets. RMSE, and standard error is reported (see §A.5 for details). We do not average over the MMLU-PROF dataset.

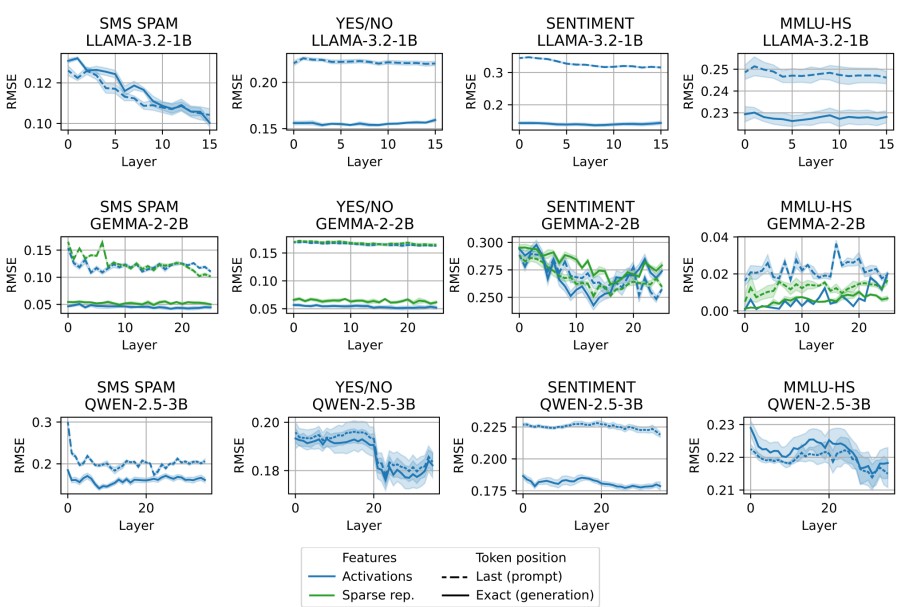

*Figure 9.* Layer-wise performance of linear error estimators for different LM, separated by distinct token position, and representation strategies.

Figure 11 shows the probe results for the logistic regression probes, used in §6. AUCROC, and standard error is reported (see §A.5 for details). Higher is better.

### A.6.2. TRANSITIONS

We report transitions in model predictions, before, and after steering using $2 \times 2$ matrices, with $\delta = 0.00$, *i.e.* without safety constraints. Each cell corresponds to a transition between binary outcome classes, incorrect (0), and correct (1), and displays

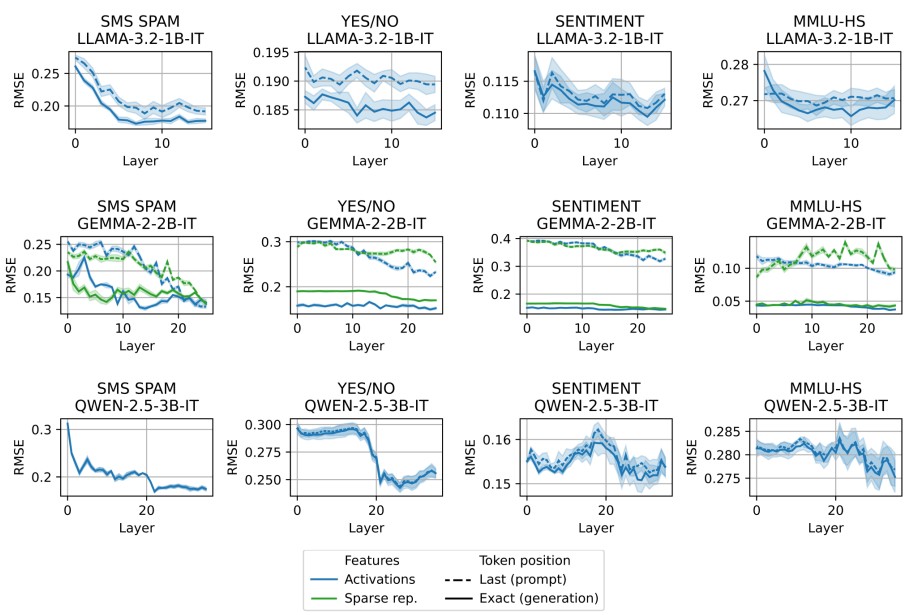

*Figure 10.* Layer-wise performance of linear error estimators for different LM, separated by distinct token position, and representation strategies.

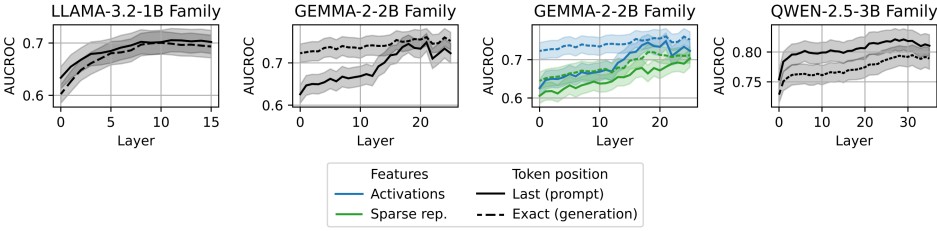

*Figure 11.* Layer-wise performance of linear error estimators for distinct LM families, separated by distinct token position, and input feature strategies. AUCROC, and standard error is reported.

counts in both the *"last"* (left), and *"exact"* (right) evaluation modes, reflecting distinct prediction outcomes after steering. Higher values in top-right cell is prefered.

As seen in Figure 12, MERA-based methods consistently induce a greater number of favourable transitions $(0 \rightarrow 1)$, while maintaining relatively low rates of degradation $(1 \rightarrow 0)$. Baseline methods, by contrast, exhibit more heterogeneous behaviour, often resulting in inconsistent effects across datasets. These results are averaged over the LMs.

### A.6.3. SPARSE STEERING

Figure 13 illustrates the sparsity of features employed during steering across different layers, and datasets. On average, only 3-5% of the latent features are activated for steering, with notable variations observed for specific tasks like MMLU-HS, where certain layers (e.g., around 15-25) exhibit lower sparsity.

### A.6.4. ABLATION STUDY

The aggregated ablation study in Figure 14 demonstrates that steering on all token positions, $\mathbf{a}_{1:(m+n)}$, and across all layers, $\ell \in \{1, \ldots, L\}$, consistently provides stronger empirical results compared to steering on selected tokens or the best layer, as identified by the highest performing probe. Specifically, the steering strategy on all tokens surpasses steering restrictions on "generation tokens", *i.e.* $\mathbf{a}_{k>n}$ in improving task accuracy. Similarly, applying interventions across all $L$ layers yields higher overall performance compared to limiting interventions to the "best layer".

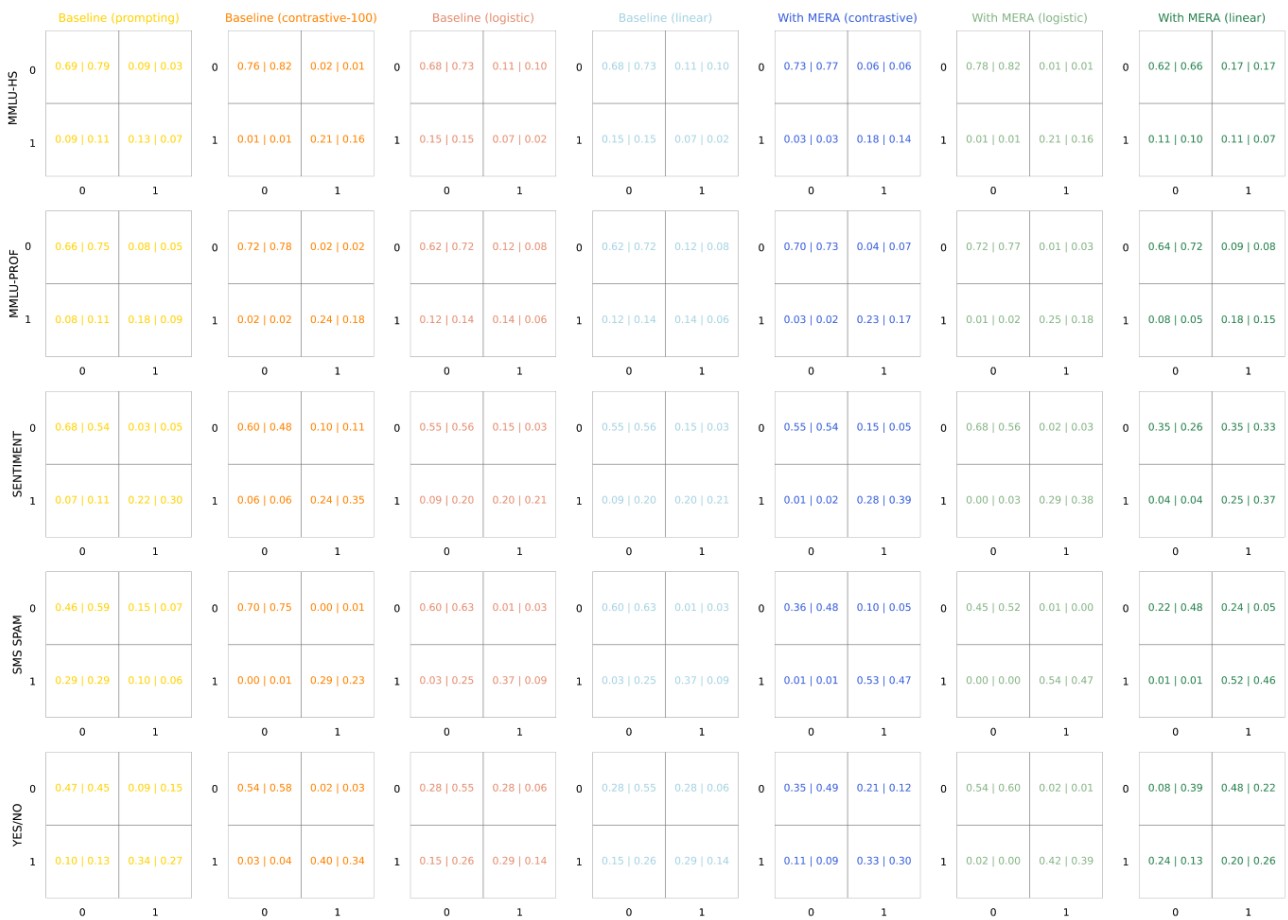

*Figure 12.* Transition matrices for each dataset, and method pair. Each cell quantifies the number of predictions transitioning from one correctness state to another after steering: $0 \to 0$, $0 \to 1$, $1 \to 0$, and $1 \to 1$, where 0 denotes incorrect, and 1 denotes correct. Values are shown for both the *"last"* (left), and *"exact"* (right) evaluation modes.

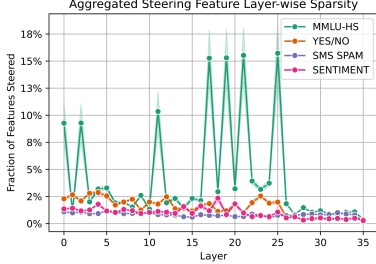

*Figure 13.* With probe steering, on average 3-5% of the latent features are employed for steering.

**Token Position-wise Probe Performance**    As shown in Figure 15, the probe trained for a single token position demonstrates consistently that the average RMSE is comparable across other token positions, including those not explicitly targeted during training. Naturally, some variability is expected. That said, these results suggest that the probe generalises sufficiently well to the remaining token positions retaining performance close to that of directly generated model tokens. For this analysis, we selected four random combinations of models, and datasets.

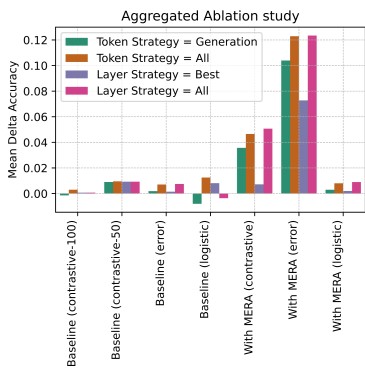

*Figure 14.* Ablation study, aggregated over all datasets but the MMLU-PROF dataset, and models.

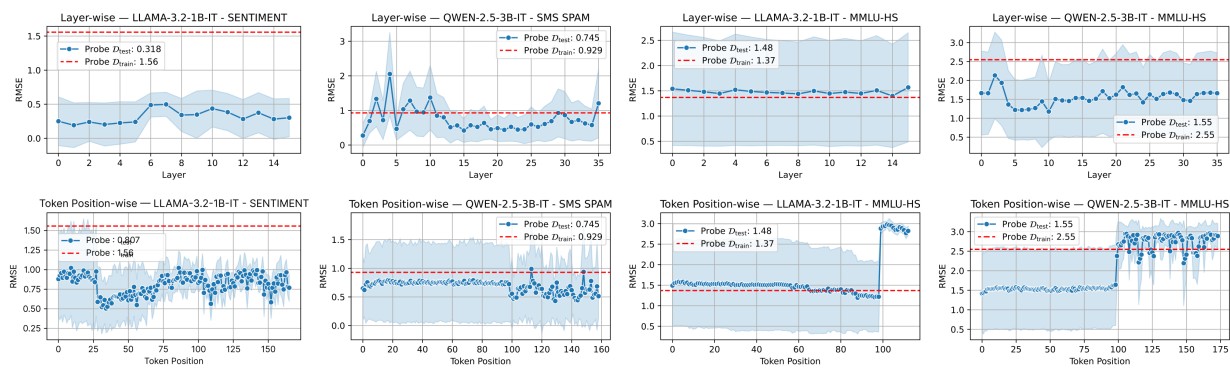

*Figure 15.* Comparison of probe performance across all token positions, and layers, including those not explicitly targeted during training.

## A.7. Accuracy vs Error

The plot in Figure 16 illustrates the relationship between Delta Test Accuracy (↑), and Delta Test Error (↓) across various models. Each point represents a different steering method or baseline, highlighting the relationship between improving accuracy, and reducing errors. The clustering along a diagonal trend suggests a strong negative correlation.

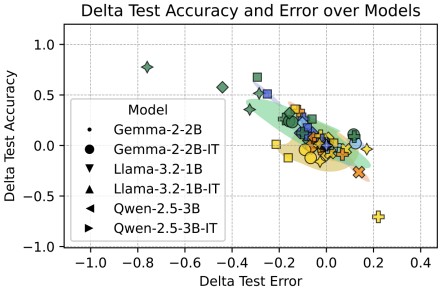

*Figure 16.* Delta Test Accuracy (↑) versus Delta Test Error (↓) across the different models, over the steering methods, and baselines.

