# OpenReview forum: "To Steer or Not to Steer? Mechanistic Error Reduction with Abstention for Language Models"
_ICML.cc/2025/Conference — ICML 2025 poster_

### Official Review · Reviewer_MXKB · 2025-03-13

**Overall Recommendation:** 3

**Summary:**

This paper introduces Mechanistic Error Reduction with Abstention (MERA), a framework for conditional activation LM activation steering that addresses a fundamental challenge in the steering literature: that interventions can often hurt overall performance and are often applied unnecessarily. Unlike traditional steering methods that apply fixed intervention strengths, MERA trains linear error estimators to predict model errors from activations and makes calibrated steering decisions - intervening only when the predicted error exceeds a threshold and with strength proportional to the estimated error magnitude.

The work expands upon existing steering literature by focusing on continuous error mitigation for improving classification performance rather than binary alignment goals like reducing toxicity prevalent in the steering literature. The authors include experiments with multiple classification tasks and compare against popular baseline methods.

**Claims And Evidence:**

This work claims that MERA is an improvement over baseline steering techniques. While MERA does seem to generally outperform baseline steering techniques, this claim can benefit from additional clarification.

**Inconsistent Gains**: Line 361 notes that MMLU-HS does not meaningfully benefit from MERA. The authors note that the increased class cardinality over the other binary classification tasks may make it an especially difficult setting to steer. That MERA struggles to generalize beyond binary classification tasks suggests that the technique may suffer from challenges in generalization faced by baseline techniques (line 81).

**MERA May Still Be Expensive**: The paper implies that MERA is more practical than baseline techniques since baselines often require expensive hyperparameter searches (Lines 199 & 416). However, MERA's multi-step approach, including training an auxiliary model as well as searching for optimal hyperparameters (Line 300), suggests that MERA is not an obvious efficiency improvement over baseline techniques when it comes to hyperparameter optimization.

**Essential References Not Discussed:**

There are no essential references which wouldn't be considered concurrent work as far as I can tell.

**Experimental Designs Or Analyses:**

See previous section

**Methods And Evaluation Criteria:**

**Benchmarks**: The studied benchmarks overall make sense for studying simple classification tasks. However, it is unclear why only the High School splits of MMLU were included. Including all MMLU splits would give readers a sense as to how well MERA generalizes to more difficult topics.

**No Steering Results in Main Paper**: The main paper does not include the absolute baseline results. The main results instead include a bespoke metric with relative improvements in performance. While this bespoke metric can tell readers whether accuracy improves or regresses, it is unclear to what absolute degree MERA improves benchmark performance.

**Other Comments Or Suggestions:**

The citation on line 671 appears broken.

**Other Strengths And Weaknesses:**

Safety-related behavior is a common focus of the steering literature. This paper focuses on steering for error reduction on classification tasks, which is a differentiator and a strength. The paper also raises the interesting point that this setting, where there is a verifiable correct answer, is a practical test-bed for steering techniques.

**Questions For Authors:**

Can the authors please clarify how their approach to conditional steering differs and or improves upon [1]?

Lee, B.W., Padhi, I., Ramamurthy, K.N., Miehling, E., Dognin, P.L., Nagireddy, M., & Dhurandhar, A. (2024). Programming Refusal with Conditional Activation Steering. ArXiv, abs/2409.05907.

**Relation To Broader Scientific Literature:**

Activation steering has emerged as an alternative method to prompting for dynamically controlling LM behavior at inference time. While great progress has been made in eliciting behavior of interest via steering, adverse effects on overall performance remain one of the primary open problems in the field. This paper introduces a conditional steering technique for sidestepping performance issues by estimating the LM's error ahead of time and only steering if the situation calls for it. This is a timely and relevant contribution to the activation steering literature.nd only steering if the situation calls for it. This is a timely and relevant contribution to the activaiton steering literature.

**Theoretical Claims:**

NA

---

> ### Author Rebuttal · Authors · 2025-04-01
>
> Thank you for all these important remarks. We’re encouraged that you found our work a timely and relevant contribution to the steering literature and that our focus on steering for error reduction is a differentiator/ strength! We’ve addressed all of your remarks below. Please let us know of any remaining questions.
>
> **1) Inconsistent gains.** We also note that MMLU-HS is more difficult to steer than the other tasks evaluated (except Llama-IT!). This corroborates [3] + aligns with a broader limitation shared by all additive linear steering methods — assuming that the activations and the target quantity (here, the model error) are linearly related. One of MERA’s key contributions is to explicitly detect and gracefully handle these “unsteerable” instance cases or datasets. Our selective steering mechanism (see Eq. 7) and then also the abstention criterion (see Sec. 3.2), ensure that we intervene only when confident (which is controlled by the parameter $\delta$). Rather than steering on all inputs and datasets, MERA steers conditionally.
>
> On reflection, we see that attributing MMLU's limited steerability only to class cardinality is premature. Other confounding factors such as label ambiguity, semantic class overlap, or general task complexity may be equally or more relevant. It may be that steering toward correct answers might be easier when classes lie along a single semantic axis (e.g., yes/no, positive/negative) than in a general MCQA format. This needs to be empirically verified — characterising steering techniques’ limits wrt distinct dataset properties deserves careful consideration and is an exciting direction for future work (see e.g., [1]). We've now removed the prior formulation citing cardinality as the primary cause for MMLU-HS’s limited steerability. We appreciate your attention to this detail and the opportunity to improve our paper in this regard!
>
> **2) MERA may still be expensive.** You're right that MERA introduces its own components, such as probe training and calibration. However, these steps are principled and crucially, MERA avoids per-model*, per-layer, per-token and per-multiplier hyperparameter sweeps, which is the common sweep strategy in activation steering**. This creates a huge search space and brings on huge computational costs, typically forcing the practitioner to prioritise one hyperparameter over the other (we refer to Appendix A.1 for a discussion, which we’re also currently expanding). MERA replaces this process with a search of ~10 $\alpha$ values per task (dataset and model combination). This seriously reduces computational costs and as we see it, establishes MERA as a more efficient method in comparison.
>
> *Even with significant compute, the best outcome of steering papers is often model-specific advice like “steer on layer 13 for Llama-7B” which may not generalise well.
>
> **An Anthropic blog post [9] summarises current steering practice well: "For all evaluations, we varied the 'steering factor' between -20 and 20. This decision was arbitrary."
>
> **3) Benchmarks.** We used the MMLU-HS subset simply because the questions would be consistent in difficulty and thus make per-sample comparisons cleaner. Practically, this subset also matched the sizes of the other datasets in the paper. But we agree that it would be interesting to evaluate MERA on more difficult subsets of MMLU. Consequently, we're currently running the MMLU "professional" subset as well.
>
> **4) No steering results in main paper.** As steering is a post-training intervention, we found it more important to focus on reporting the _relative_ steering performance in the main manuscript (over the absolute results) but we appreciate that absolute metrics provide complementary information that can be helpful to understand true steering effects! In Table 4 in the Appendix we included all the unsteered accuracies for each model and dataset combination (from which steering gains could be indirectly judged) but to make it simpler for the reader, we’ll complement this table with an additional table containing steered accuracies as well as the raw deltas. We will discuss these results in the main manuscript as well. Also, to understand why a bespoke metric was introduced, see answer (3) of Reviewer u9yC.
>
> **5) Code.**  Please see answer (3) of Reviewer 7dxY.
>
> **6) Broken citation.** Line 671 is fixed.
>
> **7) Difference to CAST.** Thank you for sharing this relevant work! We have read the paper (and cite it in Appendix A.1). Our method differs from CAST in several ways (i) conditioning logic: MERA conditions on predicted error and CAST on cosine similarity between “condition vectors” and the model’s activations exceeding $\theta$, (ii) steering direction: MERA learns probe weights per layer and CAST uses PCA on contrastive pairs, with multiple directions, (iii) strength selection: MERA solves for $\lambda^{\star}$ with closed form and CAST uses a grid search.
>
> [8] https://www.anthropic.com/research/evaluating-feature-steering

---

> > ### Comment · Reviewer_MXKB · 2025-04-02
> >
> > Thank you for the thorough response to my suggestions. My primary open concern is missing MMLU data. Do the authors expect to have the remaining MMLU experiment results available before the end of the discussion period?
> >
> > Update (April 7th): I have raised my score in response to the additional MMLU experiments

---

> > > ### Author Response · Authors · 2025-04-07
> > >
> > > Thank you for responding and sharing your remaining concern! That's really helpful. We appreciate the opportunity to provide these additional MMLU results.
> > >
> > > Below we report both the raw unsteered and steered accuracies on the professional MMLU subset across the six language models. We also include four common baselines: steer with prompt (adding a suffix: "Think before you answer"), with additive probe and with contrastive (50, 100) pairs. The table reports accuracy in "last" and "exact" prediction modes.
> > >
> > > **Table: MMLU-Professional – Accuracy (Delta)**
> > >
> > > | Model            | Mode  | Unsteered | With Prompting     | With Additive Probe   | With Contrastive-50   | With Contrastive-100   | With MERA         |
> > > |------------------|-------|-----------|---------------------|------------------------|------------------------|-------------------------|-------------------|
> > > | Llama-3.2-1B     | Last  | 0.252     | 0.257 (+0.01)       | 0.238 (-0.01)          | 0.233 (-0.02)          | 0.224 (-0.03)           | 0.262 (+0.01)     |
> > > |                  | Exact | 0.233     | 0.086 (-0.15)       | 0.219 (-0.01)          | 0.229 (-0.00)          | 0.210 (-0.02)           | 0.262 (+0.03)     |
> > > | Llama-3.2-1B-IT  | Last  | 0.295     | 0.248 (-0.05)       | 0.300 (+0.01)          | 0.257 (-0.04)          | 0.252 (-0.04)           | 0.295 (+0.00)     |
> > > |                  | Exact | 0.262     | 0.152 (-0.11)       | 0.052 (-0.21)          | 0.248 (-0.01)          | 0.271 (+0.01)           | 0.262 (+0.00)     |
> > > | Gemma-2-2B       | Last  | 0.252     | 0.252 (+0.00)       | 0.252 (+0.00)          | 0.252 (+0.00)          | 0.252 (+0.00)           | 0.252 (+0.00)     |
> > > |                  | Exact | 0.000     | 0.010 (+0.01)       | 0.000 (+0.00)          | 0.019 (+0.02)          | 0.005 (+0.01)           | 0.067 (+0.07)     |
> > > | Gemma-2-2B-IT    | Last  | 0.248     | 0.271 (+0.02)       | 0.252 (+0.00)          | 0.305 (+0.06)          | 0.262 (+0.01)           | 0.248 (+0.00)     |
> > > |                  | Exact | 0.190     | 0.195 (+0.01)       | 0.205 (+0.02)          | 0.224 (+0.03)          | 0.167 (-0.02)           | 0.195 (+0.01)     |
> > > | Qwen2.5-3B       | Last  | 0.333     | 0.338 (+0.01)       | 0.252 (-0.08)          | 0.319 (-0.01)          | 0.310 (-0.02)           | 0.333 (+0.00)     |
> > > |                  | Exact | 0.310     | 0.262 (-0.05)       | 0.233 (-0.08)          | 0.290 (-0.02)          | 0.271 (-0.04)           | 0.310 (+0.00)     |
> > > | Qwen2.5-3B-IT    | Last  | 0.190     | 0.195 (+0.01)       | 0.190 (+0.00)          | 0.271 (+0.08)          | 0.271 (+0.08)           | 0.271 (+0.08)     |
> > > |                  | Exact | 0.190     | 0.195 (+0.01)       | 0.190 (+0.00)          | 0.271 (+0.08)          | 0.271 (+0.08)           | 0.195 (+0.01)     |
> > > ---
> > >
> > > Some key observations:
> > > - MERA is safe — matches or improve accuracy in all cases (at worst no improvement, but at best 8% improvement in accuracy in the exact match which measures accuracy in the real generated model answer).
> > > - Baselines like the contrastive methods, additive probes and prompting are more variable — sometimes helpful, sometimes quite degrading in performance (see e.g., Llama-3.2-1B and Llama-3.2-1B-IT with sharp drops)
> > >
> > > These results are supporting our already reported findings in the main paper. Our view from this is of course that **steering should not always be applied**. The lower the value of $\delta$, the more MERA would abstain and subsequently be more safe. This is unlike baselines, which apply the same fixed-strength intervention to all inputs.
> > >
> > > **Changes to the paper.** We highlighted the absolute accuracies here, but we will of course integrate all the additional results to the paper: adding raw accuracy/error tables (like above) + deltas, updating SPI in Table 1, and also, importantly, expanding our discussion with an additional paragraph more directly discussing the limitations of additive linear steering, echoing our previous response to your reivew. Lastly, we’re also exploring **nonlinear extensions** of MERA as discussed in Reviewer H7hE answer (6) to advance MERA. We believe non-linear probe-based steering can offer more expressive yet still safe (non-degrading) interventions.
> > >
> > > We’d be happy to clarify anything further. It is to our hope that this additional evidence helps resolve your concern and could justify raising the score.

---

### Official Review · Reviewer_u9yC · 2025-03-13

**Overall Recommendation:** 4

**Summary:**

The authors a new steering technique (MERA). MERA formulates steering as an error reduction problem. It first trains a linear probe to determine the linear direction which is most effective for reducing the error. It then adaptively selects a steering multiplier alpha based on how far the prediction is from the desired threshold.

Unlike previous methods, which use a global fixed alpha, MERA chooses alpha adaptively on a sample-wise and token-wise basis. Compared to previous work, MERA solves the problem of under-steering or over-steering specific examples. As a result, if the prediction was already correct, MERA avoids doing steering.

The authors compare to prior work (contrastive activation addition and probe-based steering). They also ablate different parts of their methodology (MERA baselines). They evaluate on four different binary tasks, and claim MERA generally outperforms baselines.

## Update after rebuttal

The authors have improved the paper substantially with additional experiments, as well as improved the quality of writing. I have updated my score to a 4.

**Claims And Evidence:**

The authors claim that existing methods may 'over-steer' or 'under-steer', and use this as a motivating factor for MERA. It is not clear to what extent this happens with existing methods. To demonstrate this more clearly, the authors should consider doing UMAP visualizations of positive and negative activations before and after steering.

The authors compare to previous methods like CAA. However, CAA requires a hyperparameter sweep to determine the optimal layer for steering. It is also apparent that the authors did perform this layer sweep for MERA and related baselines. Hence I am concerned that the baseline is weak. I think the authors should publish their layer sweep curves for CAA.

The authors base all their evaluation on SPI. It is difficult to understand what this metric measures and I would appreciate a more in-depth explanation with figures. I would also appreciate an explanation of why other, simpler, metrics are flawed.

The authors also claim that MERA outperforms baselines. I agree with this, but would add some caveats - the models considered are mostly quite small and the tasks considered seem relatively simple.

The authors claim MERA can trade off safety and efficiency (fig 5). I did not understand this; what is delta and where did it come from? Why is (1-delta) interpreted as confidence? This claim could be explained much better.

The authors mention in Discussion that "LMs are frustratingly error prone". It is unclear what evidence was provided to support this claim - did the authors evaluate the models with no steering on the tasks provided?

**Essential References Not Discussed:**

Not as far as I am aware.

**Experimental Designs Or Analyses:**

Yes, I checked the experimental design and analysis. Please refer to 'claims and evidence' above.

**Methods And Evaluation Criteria:**

The proposed steering method makes sense. The evaluation benchmarks and metrics also broadly make sense.

**Other Comments Or Suggestions:**

I think Fig 3 (overview of MERA) should be prominently displayed on page 2.

There are many mathematical quantities defined throughout the paper. It would help to have a summary table of definitions and interpretations (preferably co-located with Fig 3).

Generally the writing contains many dense mathematical equations. It would help the reader to unpack these equations and make them easier to understand. Plausibly, it is not important to have these equations in the main paper at all - the interpretations seem more important.

**Other Strengths And Weaknesses:**

It is currently unclear to what extent the authors' findings generalise beyond the setting considered in this paper. Currently, the authors primarily consider rather small LLMs and rather simple classification tasks. Conceptually, the method seems hard to generalise to free-form generation, since it involves training linear probes.

**Questions For Authors:**

Did you do any experiments on larger models?
Did you try any free-form generations? Do the answers make sense when you do that?
How did you convince yourself that current methods 'over-steer' or 'under-steer'?

**Relation To Broader Scientific Literature:**

Better methods for steering are valuable as they facilitate few-shot adaptation; this has benefits for transfer learning and personalization. The method seems directly applicable to small LLMs run on local, consumer-grade hardware for specific tasks. More generally, the mathematical formulation provided may become a building block in future work.

**Theoretical Claims:**

The authors claim their steering method precisely solves the error reduction problem. I did not rigorously check the proof, but the broad argument makes sense.

---

> ### Author Rebuttal · Authors · 2025-04-01
>
> Thank you for the very detail-oriented and helpful review! We’re excited that you believe our work has benefits for transfer learning and personalisation and that our mathematical framework may become a building block in future work. We have addressed all of your points below:
>
> **1) Evidence for over-/under- steering.** We appreciate the suggestion — thank you for the suggestion—visualising activation space geometry (as in [4, 6]) can indeed help interpretation. However, instead of using UMAP, we now provide a more direct analysis of over- and under-steering effects by reporting transition counts on a sample-by-sample basis: how many examples move from incorrect to correct (0→1), correct to incorrect (1→0), or remain incorrect (0→0) or correct (1→1). This analysis will be included in the Appendix together with the SPI results and referenced in the main manuscript.
>
> **2) Baseline layer sweep.** Please see the answer (2) in Reviewer H7hE.
>
> **3) Evaluation with SPI.** We identified the need to define a new metric (SPI) since we ourselves couldn’t find a suitable metric that not only expresses improvement but critically, also _degradation_ in a bounded, interpretable way. Existing metrics (e.g. from OpenAI [7]) only track positive gains. By normalising relative to performing ceiling (or floor), the reader can get a sense of how _good_ the steering method is compared to how its best or worst case. How much (in percentage %) does steering improve or degrade LM performance? If SPI=1, steering makes the LM a perfect classifier. If SPI=-1, steering made the LM incorrect on all test samples. SPI is easy to interpret and particularly helpful when comparing LMs side-by-side (which often differ vastly in initial unsteered performance!). We’ll make sure to add an expanded, more clear motivation of the metric in the paper.
>
> **4) Caveats on scope.** Totally fair — our experimental scope includes language models going up to 3B and datasets of multi-choice. Given our compute constraints, we prioritised coverage across model types (base vs. instruction-tuned, and three distinct families), and chose supervised tasks to allow controlled evaluation where error can be perfectly recovered at distinct token positions (and not approximated e.g., using external oracle LMs). We'll emphasise our scope better in Section 7.
>
> **5) Clarifying $\delta$.** In our method, $\delta$ is user-defined and sets the confidence level for steering. For instance, setting $\delta$ = 0.05 corresponds to a 95% confidence threshold: steering is applied only when we're statistically confident that it will improve performance. To make room for new clarifications from this rebuttal, we’ve decided to remove this section in the revised manuscript! Thank you again!
>
> **6) LMs error proneness.** Thank you for flagging this — we’ve adjusted the wording to “[can be] frustratingly…” instead of “[are] frustratingly…” to better reflect variability. As shown in Table 4 (Appendix), even capable base/ IT models can sometimes yield low accuracy (e.g., 5.6%) and high error rates such as 0.89) on certain tasks.
>
> **7) Clarifying generalisability.** While we focus on classification tasks for clarity and control, MERA is a general framework. It only requires a real-valued or binary signal to train the probe — so it can, in principle, be applied to other supervision types like truthfulness, toxicity, or helpfulness! Our optimisation and calibration steps are agnostic to task type and independent of model size. To apply MERA to free-form generation, each output should be paired with a target label. This can be done, for instance, using external oracles or LM-based rating systems. We’ll clarify this in Section 7 to better highlight MERA’s broader applicability.
>
> **8) Formatting.** We have revised Figure 3 for improved clarity! We have also streamlined the writing and notation in Sections 2 and 3 to make the mathematical content accessible.
>
> **9) Questions.** a) We have not yet evaluated any _larger_ models than 3 billion parameters, but are planning to do that with access to more compute. b) Yes — in all our experiments, the steering methods are evaluated in two complementary modes: "last" and "exact". (i) In "last", we evaluate the model’s logits at the final prompt token. (ii) In "exact", we check whether the model generates the correct answer in its free-form output… where the second “exact” mode accommodates the open-ended settings! c) Fixed-strength baselines apply the same intervention regardless of the model’s current activation state, making over- or under-steering inevitable by design. Empirically, we observed this in two ways: (i) that baselines often output negative SPIs (i.e., steering inadvertently degrades the model), and (ii) instance-level transitions where steering flips correct predictions to incorrect ones. We’ll now include the transition analysis in the paper!
>
> [6] https://arxiv.org/pdf/2312.01037
>
> [7] https://openai.com/index/weak-to-strong-generalization/

---

> > ### Comment · Reviewer_u9yC · 2025-04-01
> >
> > Thank you for your detailed response. I am willing to upgrade my score to a 4 conditioned on the stated improvements being made.

---

> > > ### Author Response · Authors · 2025-04-07
> > >
> > > Thanks again for your comments + for being open to raising the score.
> > >
> > > Just to confirm — we're currently rerunning the full benchmarking to expand scope with transition analysis (0→1, 1→0, 0→0, 1→1) across all 6 language models and 5 datasets (we also added an additional MMLU subset, see our recent rebuttal response to Reviewer MXKB). Results so far align well with our reported SPI trends. We'll also add absolute steered accuracies and the differences to unsteered accuracies in the Appendix. All the other points you flagged such as improving description and motivation of SPI, being more clear on scope and MERA's generalisability and improving wording/ formatting etc are also addressed.

---

### Official Review · Reviewer_H7hE · 2025-03-14

**Overall Recommendation:** 3

**Summary:**

The paper proposes a new latent space steering methodology for LLMs. The basic setup is that we  prompt an LLM with a question from a finite-class classification task, and we let it generate an open-ended response. We can decode the model's prediction from its open-ended generation by either
- **last token**: using the argmax over discrete labels of the logits at the last token of the prompt; "last" is a bit of a misnomer, since really it's the last token of the prompt.
- **exact token**: find the first match to one of the discrete labels in the generation, and use that as the prediction (together with its associated next-token probability given by the model)

The **error** of the model, denoted $E(z)$ in the paper, is a soft measure of how far the generation is from the true answer, and is measured as 1 minus the probability of the true answer (under either "last token" or "exact token"). The goal with steering is to make the model better at the classification task, i.e. increase accuracy. $1-E(z)$ is a soft proxy for the accuracy.

The main innovations of the steering method are:
- it frames the steering problem as a constrained optimization, minimizing the magnitude of the latent space perturbation subject to achieving a certain degree of desired effect on $E(z)$, measured via a proxy, namely a linear probe on the activations that is intended to predict the $E(z)$.
- In particular, when the unperturbed activation already has a satisfactorily low error, it is unchanged. Thus the steering is conditional
- it also additionally frames the degree of desired effect to seek in steering as another optimization problem, rooted in a more "human-interpretable" measure, e.g. the 0-1 based, "hard" metric of accuracy.

The method is evaluated on several classification datasets and several LLMs in the parameter range 1B-3B, both base and instruction-tuned. The choice of steering direction (probe vs other options) is ablated. Baselines such as contrastive steering via difference in means are considered. Results show mostly improvement over baselines.

Some problems with the methodology, correctness and analysis are discussed in subsequent sections of this review.

## Update after rebuttal
The rebuttal has not meaningfully changed my recommendation; I am still in favor of accepting the paper if possible.

**Claims And Evidence:**

The claims are largely supported by the given evidence. The most informative artifact is Table 1, which shows performance of all methods by both LLM and dataset. The claims are quite straightforward, clear & verifiable by looking at the metrics reported.

**Essential References Not Discussed:**

Given that framing steering as an optimization problem is central to the message of this paper, it would benefit from a brief comparison with the below paper which takes a very different optimization approach:

Cao, Y., Zhang, T., Cao, B., Yin, Z., Lin, L., Ma, F. and Chen, J., 2024. Personalized steering of large language models: Versatile steering vectors through bi-directional preference optimization. _Advances in Neural Information Processing Systems_, _37_, pp.49519-49551.

**Experimental Designs Or Analyses:**

- The designs are clear and straightforward.
- The plots in Figure 4 combine across datasets and/or models in a way quite prone to noisy results and unclear conclusions. Table 1 shows significant variation across these conditions. E.g., MMLU barely shows any improvement at all, while the SMS SPAM dataset shows huge improvement. This is further complicated by the fact that the metric SPI being averaged is also difficult to interpret in the absence of the accuracy of a given model on a given dataset. Averaging these wildly different values likely reduces these "aggregates" to metrics dominated by noise and/or the conditions exhibiting the strongest effects.

**Methods And Evaluation Criteria:**

Some issues:
- the linear probe to estimate error from activations has as its targets the error **probability**, a quantity in the range $[0, 1]$ with quite nonlinear behavior (going from 0.90 to 0.95 reflects a much more significant change in model internals than from e.g. 0.45 to 0.5). The regression loss is mean squared error. This doesn't quite typecheck: it is advisable to use the pre-softmax logits as the targets of regression instead. These quantities are "linear-ish" functions of internal activations with unbounded range where relative changes at different levels are more comparable.
- I am confused by the description of the baseline BASE-$\mu_k$.
	- Why do we use for this baseline the layer "identified as most effective by a probe"? Shouldn't we use the layer in which this baseline itself is most effective, regardless of what other methods might tell us? It feels like mixing multiple methods to bias the baseline instead of just getting the true performance we can squeeze from the baseline (of course, these might happen to be the same. But this is a methodological issue).
	- Second, it is unclear how the choice of $k$ for the number of contrast pairs to use was made. This is important as there are cases in Table 1 where 50 vs 100 leads to dramatic differences. I get that we want extreme correct/incorrect examples to get a contrastive direction here, but still

**Other Comments Or Suggestions:**

- writing is not very clear in 3.2, 3.3 - there are many questions left, such as:
	- what is meant by "This direction is then scaled using the closed-form solution at both the token and layer levels"?

**Other Strengths And Weaknesses:**

Strengths:
- I think that framing steering as an optimization problem is a great way to move the fields towards more principled foundations, and to be clear and precise about what we are trying to achieve and why.
Weaknesses:
- While the method is mostly agnostic to the steering target, the motivation for the particular choice of steering objective is quite questionable. Why would we hope that steering - a blunt and extremely low-expressiveness instrument - would be able to make a language model "smarter"? In a strong sense, steering can only "work with what is already there", surfacing and amplifying existing representations. Intuitively, the only way for this to work is if the model has a **systematic** failure mode on some dataset, and furthermore, this systematicity is somehow represented internally in a crisp way. For instance, consider some previous work cited below that showed that you can do steering interventions on an LLM to make it more truthful on the TruthfulQA dataset. This dataset contains many questions that have common misconceptions associated with them. It makes sense that an LLM would represent internally both the correct answer (which is also consistently represented in the pretraining data), as well as the widely repeated incorrect one (being a next-token predictor). Thus it makes sense that we can steer the model towards saying the less common answer. However, for a dataset like MMLU, it would be extremely surprising if we can get strong gains from just steering. This begs the question: what do steering experiments on MMLU really teach us?



Li, K., Patel, O., Viégas, F., Pfister, H. and Wattenberg, M., 2023. Inference-time intervention: Eliciting truthful answers from a language model. _Advances in Neural Information Processing Systems_, _36_, pp.41451-41530.

**Questions For Authors:**

N/A

**Relation To Broader Scientific Literature:**

I think the paper did a good job at summarizing the state of steering and situating the findings within it.

**Theoretical Claims:**

The derivation of the correction term $\sqrt{\log(2/\delta) / (2N)}$ for the calibration threshold (3.2. Calibrating for Safety) is wrong. This is because in the sweep over values of $\alpha\in(0,1)$, we use the same random sample $D_{cal}$ multiple times. We should either sample a new i.i.d. calibration dataset for each value of $\alpha$ we try, or account for looking at the same data multiple times, which is easiest via a union bound. As a result, a reported level of confidence, e.g. 95%, should be treated as a lower level of confidence; with 10 values of $\alpha$, it could be as low as $100 - (10*5) = 50$ percent via a union bound.

---

> ### Author Rebuttal · Authors · 2025-04-01
>
> Thank you for all the time taking to provide a very detail-oriented and helpful review! We’re glad to hear that you found our claims straightforward, clear + verifiable, our optimisation framing is a great way to move the fields towards more principled foundations and our experimental designs are clear. Now, we have addressed all of your points below:
>
> **1) Regression target for probes.** Regarding learning the linear model not directly on the error but on the inverse of the sigmoid (i.e., the logits), we agree that this is a promising direction that could potentially enhance the performance of MERA. We are currently re-running this experiment on all evaluations to assess its impact. Preliminary results steering Llama and Gemma models on SMS spam dataset suggest a slight improvement over the original formulation. We’ll reassess when the complete results are in and if supported, update the methodology accordingly.
>
> **2) Describing BASE-$\mu_{k}$.** Thank you for pointing this out! We agree that mixing strategies across methods can undermine fair comparisons and we appreciate the opportunity to clarify! In our experiments, we do not use any MERA-specific best-performing layer for the BASE-$\mu_{k}$ baseline. Instead, we choose the layer with the lowest RMSE under a trained linear probe — a shared external signal reflecting where the error is most linearly related to model activations —which we think enables the fairest comparison on balance. As we see it, probe performance is a good heuristic for selecting contrastive steering layers, since it neutrally depends on how linearly our target is related to the activations (and does not favour MERA which steers on _all_ layers). It is worth mentioning that the community hasn’t converged to one single criterion for layer selection for contrastive difference-in-means steering (see distinct approaches in [2-5]) but that many methodologically valid strategies exist. We’ll revise the section to make our approach clearer!
>
> On the choice of $k$ — we agree that results vary significantly between $k = 50$ and $k = 100$. To us, this reflects a general instability in contrastive baselines. To construct these sets, we sorted the error scores of the 3000 training samples and selected the top-k and bottom-k examples. We’ll add these details to the manuscript as well!
>
> **3) Theoretical claim derivation.**  We acknowledge that there is indeed a selection bias in our original derivation, since the same calibration dataset was used both to construct the confidence intervals and to select the best-performing $\alpha$. We have now addressed this issue by adding a Bonferroni correction to the confidence bound. As a result, the reported confidence levels have been adjusted: the results previously shown at 99\% confidence are now updated to 90\%. Full details are provided in our response to Reviewer 7dxY. We also outline a less conservative alternative and will clarify in the next paper version that our updated method has formal guarantees under the i.i.d. assumption.
>
> **4) SPI aggregration.** We appreciate this observation. Figure 4 was intended to offer a practitioner-oriented overview (for a quick comparison across settings to help identify overall trends) and Table 1 to provide complete details on model- and dataset-specific effects. As we see it, these provide _complementary perspectives_ which are helpful to the reader in distinct ways. We now emphasise this dual purpose more clearly in the manuscript and explicitly caution readers about over-interpreting the aggregated view.
>
> **5) Personalised steering citation.** Thank you for pointing us to this interesting read! We find the work highly relevant as a steering reference and have thus cited it in our paper (+ added a discussion in Appendix A.1). This work is different from ours in several ways, most notably, in the specific problem it is targeting — where they don’t directly solve for steering strength ($\lambda$) (but find it via hyperparameter sweeps, see Table 5 on p.8). This contrasts with MERA, which gives the solution in a closed form.
>
> **6) Steerability and MMLU.** What MMLU can/not teach us is a very fascinating comment! We refer to answer (1) in Reviewer MXKB for a discussion on this topic.
>
> Naturally, linear steering has its limitations. In this paper, we suggest possible extensions in Appendix A (Non-linear Case). A simpler alternative, however, is to use a first-order approximation of the non-linear model. This approach would essentially replace the fixed linear weight used across all instances with the gradient of the non-linear function for each specific input.
>
> **7) Improved writing.** Thank you for taking the time to write out the example where our writing could be improved! We’re revising Sections 3.2 and 3.3 and also the rest of the paper for clarity.
>
> [2] https://openreview.net/pdf?id=HuNoNfiQqH
>
> [3] https://arxiv.org/pdf/2312.06681
>
> [4] https://arxiv.org/pdf/2310.06824v3
>
> [5] https://arxiv.org/pdf/2402.14433

---

> > ### Comment · Reviewer_H7hE · 2025-04-08
> >
> > Thank you for engaging with my review.
> >
> > Re:BASE-$\mu_k$, I'm still unconvinced this is a fair baseline. Many results in the literature show that it is possible to very successfully probe for certain concepts even in layers where altering the concept has no causal effect on model behavior. Furthermore, the choice of $k$ remains arbitrary from my point of view

---

> > > ### Author Response · Authors · 2025-04-09
> > >
> > > Thank you. We agree that your suggestion makes sense in principle. Selecting the best-performing layer per baseline and $k$ is ideal if one has the resources to do it. But, as you're likely aware, doing this properly would require a massive number of runs as all hyperparameters are entangled (see Reviewer MXKB answer (2) for a comment on this). To make the choice of the empirically best layer and $k$ combination, we’d need to run 3120 steering evaluations _per baseline_. And if we opened up token position choices (like in [5]) the combinatorics would explode. For context, if we assume we try 4 choices for $k$ then:
> > >
> > > * LLaMA-1B: 16 layers × 4 × 5 tasks = 320
> > > * Gemma-2B: 26 layers × 4 × 5 tasks = = 520
> > > * Qwen-3B: 36 layers × 4 × 5 tasks = 720
> > >  → Total = 1560 runs × 2 (both base + instruction tuned model) = 3120 runs per baseline
> > >
> > > This scale of sweeps isn’t feasible for us. What can be added here is that we did perform an ablation experiment in the Appendix where we intervened on all layers as well (not just best selected by the probe) for the contrastive baselines. We did not find any significant difference in scores between the two approaches.
> > >
> > > As you know, no single heuristic for choosing layer or $k$ is perfect: prior work has done a variety of things like using values as low as 20 or 50 ([6], [7]), pruning contrastive pairs for more signal post hoc ([8]) with different layer selection strategies ([1–4]). Our aim here was to use a methodology that is cheap + consistent across the baselines. Our probe-based approach does not directly imply casual steerability (as you point out!) but it’s simple and importantly doesn’t privilege any method.
> > >
> > > Incidentally, this is exactly the problem MERA is designed to solve. Rather than tuning over $k$ and the layers + other steering hyperparameters, MERA steers selectively but only when the predicted error is higher than a calibrated error threshold ($\alpha$). It sidesteps the need for exhaustive tuning altogether.
> > >
> > > To support a broader contrastive comparison, we will also include $k = 200$ baseline variant in the main paper.
> > >
> > > [5] https://arxiv.org/pdf/2308.10248
> > >
> > > [6] https://openreview.net/pdf?id=HuNoNfiQqH
> > >
> > > [7] https://arxiv.org/pdf/2312.06681
> > >
> > > [8] https://arxiv.org/pdf/2410.01174

---

### Official Review · Reviewer_7dxY · 2025-03-19

**Overall Recommendation:** 3

**Summary:**

Current steering methods for LM error mitigation use fixed intervention strengths, which has the risk of under-/oversteering. The paper introduces MERA, which does adaptive activation steering guided by linear error probes; the intervention thresholds (α) is calibrated via Hoeffding’s inequality. The framework further optimizes steering strength per-token/layer and abstains when unnecessary, ensuring minimal intervention. Evaluations on three LLMs show gains on binary/ternary tasks while avoiding degradation seen in fixed-strength approaches.

**Claims And Evidence:**

Yes, except the Hoeffding's inequality part as I comment in "Theoretical Claims"

**Essential References Not Discussed:**

The proposed work seems highly related to an existing paper \[1\], and therefore it is important to clarify the similarity and differences between the two.  As far as the reviewer see:

1. Motivation-wise, both the work of \[1\] and the current paper mention that existing methods rely on fixed steering strength, which leads to under/over steering.
2. The two works use different existing ways of extracting a steering *direction* (independent of the strength): \[1\] uses PCA on the normalized contrastive vectors, while this paper uses linear probes. Nevertheless, \[2\] points out that the two ways of extracting steering directions are equivalent under certain conditions.
3. Given one steering direction, both work can estimate the strength of steering directions. \[1\] does so by decomposing the activation along the steering directions, while this paper does so by finding the smallest $\\ell^2$-normed vector that sufficiently reduces the probing probability. Interestingly, equation (6) of this paper share some features with proposition 2 of \[1\]: both depends on the inner product of the steering direction and the activation, which can be viewed as achieving adaptive scaling.
4. To achieve a steering task, \[1\] in its general form uses multiple directions each corresponding to a semantic concept, and finds the steering strength for all of them with one sparse decomposition problem, while this paper focuses on one direction.
5. This paper investigated the choice of the representation to steer in section 4, while \[1\] followed a prior work.

Anyhow, I understand that papers could have similarities and what not, but it is important to place the proposed work relative to the literature to attribute what is existing and clarify what is the contribution.

\[1\] PaCE: Parsimonious Concept Engineering for Large Language Models, NeurIPS 2024\.
\[2\] The linear representation hypothesis and the geometry of large language models, ICML 2024\.

**Experimental Designs Or Analyses:**

The authors observed that datasets with high cardinality (e.g., MMLU-HS with 4 classes) are overall difficult to steer. I do not know if four choices per question is a high one, since it is common as high-school level questions, and there are tests with 6 options if not more. Even in high school, one could face open questions without choices, meaning infinite many choices. Could the authors provide some insights and comments on this?

**Methods And Evaluation Criteria:**

Yes, the proposed methods make sense for the problem

**Other Comments Or Suggestions:**

N/A

**Other Strengths And Weaknesses:**

N/A

**Questions For Authors:**

N/A

**Relation To Broader Scientific Literature:**

Please find my comments in "Essential References Not Discussed".

**Theoretical Claims:**

I understand Hoeffding’s inequality and in general high-dimensional statistics, but I do not get what is the rationale of using it in line 195 (left). It would be great if one could start with Hoeffding’s inequality, explain what are random variables, which assumptions of Hoeffding’s do they (approximately) satisfy or not satisfy, and how it leads to the bound in line 195 (left).

---

> ### Author Rebuttal · Authors · 2025-04-01
>
> Thank you for the constructive and informative review! We have addressed your four key points below:
>
> **1) Hoeffing inequality — Clarifying Our Selection Procedure.**
>
> Our selection procedure ensures we do **not** choose any $\alpha \in (0,1)$ unless it **statistically significantly** improves performance. Specifically, we first identify a set of $\alpha$ values with provably positive performance gain (with high probability), then select the empirically best among them.
>
> Let $f(\alpha, X_i) \in [0,1]$ be a performance function and $X_i$ a random input. Given a calibration dataset $D_n = \\{ X_1, \dots, X_n \\}$, the empirical performance is:
>
> $$
> P(\alpha, D_n) = \frac{1}{n} \sum_{i=1}^n f(\alpha, X_i),
> $$
>
> with $P(\alpha, D)$ denoting the true (population) performance.
>
> #### Procedure Overview
>
> 1. **Discretize** $[0,1]$ into $M$ values: $\alpha_{set} = \\{\alpha_1, \ldots, \alpha_M\\}$.
>
> 2. **Confidence Bands**: Apply Hoeffding’s inequality to each $\alpha_j$:
>
>    $$
>    \Pr\left( \left|P(\alpha_j, D_n) - P(\alpha_j, D)\right| \le \delta_n \right) \ge 1 - \frac{\alpha}{M},
>    $$
>
>    where $\delta_n = \sqrt{\frac{\ln(2M/\alpha)}{2n}}$.
>
> 3. **Union Bound**: Ensures all bounds hold simultaneously with high probability:
>
>    $$
>    \left|P(\alpha_j, D_n) - P(\alpha_j, D)\right| \le \delta_n \quad \forall j=1,\ldots,M.
>    $$
>
> This yields a uniform guarantee:
>
> $$
> \Pr\left(\sup_{\alpha \in \alpha_{set}} \left|P(\alpha, D_n) - P(\alpha, D)\right| \le \varepsilon_n\right) \ge 1 - \alpha.
> $$
>
> We define the **valid set**:
>
> $$
> \alpha_{\text{valid}} = \\{\alpha : P(\alpha, D_n) - \epsilon_n > 0\\},
> $$
>
> and select:
>
> $$
> \alpha^* = \arg\max_{\alpha \in \alpha_{\text{valid}}} P(\alpha, D_n).
> $$
>
> Since the confidence bands hold uniformly with high probability, we guarantee $P(\alpha, D) > 0$ for all $\alpha \in \alpha_{\text{valid}}$. Hence, $\alpha^*$ corresponds to a **true performance improvement**, or we abstain otherwise.
>
> > **Note**: We will make this guarantee more explicit in the next version of the paper.
>
> ---
>
> ### **Alternative Calibration Perspective**
>
> Instead of using a Bonferroni-style correction (which can be conservative), one could:
>
> - **Split the data** into two parts:
>   - Use one part to find the empirically optimal $\alpha^\star$.
>   - Use the other to estimate a confidence interval for $P(\alpha^\star, D)$.
>
> We accept $\alpha^\star$ only if its confidence interval lower bound exceeds a desired threshold. This avoids overly conservative bounds and allows for tighter, adaptive inference at the cost of splitting the data.
>
> **2) Cardinality versus steerability.** Thank you for raising this important point. We agree that four classes is not especially high, and in retrospect, attributing the difficulty of steering on MMLU-HS to cardinality alone is premature (dataset charactistics like semantic overlap, label ambiguity, and task complexity could also play a role in a task’s steerability). See further discussion in answer (1) at Reviewer MXKB. As we see it, a dedicated, systematic study (e.g., along the lines of [1]) that carefully varies such characteristics while analysing steering performance would be necessary to understand the limits of additive methods like MERA (+ its baselines). We've revised the formulation in the manuscript.
>
> **3) Code availability.** Apologies — we missed attaching the code at submission. We had it ready for rebuttal, but then we just learned that ICML guidelines don’t allow updates to the original submission in the discussion phase. If we’re allowed to share an anonymous link (can AC please let us know if this is acceptable) we can upload the source code, notebooks and installation guides directly there!
>
> **4) Relation to PaCE.** Thank you for pointing out the connection to PaCE! Your analysis is very insightful. While there are mathematical similarities, as you point out (i.e. adaptive intervention strengths based on the inner product of activations and steering directions), the overarching goals of PaCE and MERA are very different. The PaCE model is trained on data demonstrating ‘benign’ and ‘harmful’ concepts from some predefined dictionary, and performs interventions to suppress the harmful ones, while MERA is fundamentally concerned with error mitigation on well-defined prediction tasks. We also can’t see a direct analogue of our calibration step, which uses calibration data to identify the optimal intervention strength for error mitigation on a per-layer basis. That said, we consider the work relevant enough for inclusion in Section 3 and have also added it to the Related Works in Appendix A.1.
>
> [1] https://arxiv.org/pdf/2407.12404

---

### Decision · Program_Chairs · 2025-05-01

**Decision:**

Accept (poster)

**Comment:**

This work presents a new framework for activation steering with the objective of reducing model errors. Unlike other methods, the proposed solution is able to automatically decide when to intervene and with which strength. By doing so the method prevents under- or oversteering. Specifically this method trains linear regression estimators of the model error from activations and intervenes only when the predicted error exceeds a threshold and with a strength that is proportional to the predicted error. The empirical validation is focused on model of size 1B to 3B and to multiple choice answers (2 to 4 answers).

The problem of under- and oversteering is relevant and all reviewers recognize the importance of working on this aspect as well as the empirical evidence shown in the reported results.

An incorrect derivation of the term for the calibration in 3.2 was highlighted by one reviewer and the authors addressed this issue recognizing the mistake and correcting the confidence bound by adding Bonferroni correction term.

Multiple reviewers mentioned that the baselines could have been stronger by performing a sweep on the layer they intervene on (I believe this would have been closer to what the authors do for their own method, i.e., train 5 models for each layer and use the one with the smaller RMSE).

The biggest concerns I have are related to the lack of evidence that the method does not degrade general utility, and the lack of discussion on the additional latency at inference time that this method implies.   All metrics reported are only for the specific task on which the method is trained to reduce the error. Typically in the literature one would assess the improvement on a specific task, while monitoring the degradation of the overall utility (e.g., by measuring overall MMLU and Perplexity, for example see Section 7 of https://arxiv.org/pdf/2312.06681).

 As for the latency issue, this method, unlike most steering mechanisms, introduces potentially non-trivial additional inference cost.

Given the overall positive reviews and the interest in this field I would accept this work but expect the authors to provide the following for the camera ready:
1. Clearly state that the model after steering is meant to be used in specialized contexts but might loose general abilities OR
2. Provide experiments about generic model abilities before and after steering (e.g., MMLU).
3. Additionally, provide an analysis of the additional inference time required when steering is employed.